# *Physalia physalis*—A Source of Bioactive Collagen for the Cosmetic Industry

**DOI:** 10.3390/ijms27010033

**Published:** 2025-12-19

**Authors:** Raquel Fernandes, Cristiana Oliveira, Diana Ferreira-Sousa, Augusto Costa-Barbosa, Paula Sampaio, Luis Reis, Javier Fidalgo, Ana N. Barros, José A. Teixeira, Claudia Botelho

**Affiliations:** 1Abel Salazar Institute of Biomedical Sciences (ICBAS), University of Porto, 4050-313 Porto, Portugalluisqueirosreis@gmail.com (L.R.); 2Centre for Animal Science Studies, Institute of Science, Technology and Agroenvironment (CECA-ICETA), University of Porto, 4200-465 Porto, Portugal; 3Centre for the Research and Technology of Agro-Environmental and Biological Sciences, CITAB, Inov4Agro, University of Trás-os-Montes and Alto Douro, UTAD, Quinta de Prados, 5000-801 Vila Real, Portugal; 4CEB—Centre of Biological Engineering, University of Minho, Campus de Gualtar, 4710-057 Braga, Portugal; cristianafilipa1999@gmail.com (C.O.);; 5LABBELS—Associate Laboratory, 4710-057 Braga, Portugal; 6CBMA—Centre of Molecular and Environmental Biology, Department of Biology, University of Minho, 4710-057 Braga, Portugalpsampaio@bio.uminho.pt (P.S.); 7Mesosystem Investigação & Investimentos by Spinpark, Barco, 4805-017 Guimarães, Portugal; id@mesosystem.com; 8Associated Laboratory for Animal and Veterinary Science (AL4AnimalS), 1300-477 Lisboa, Portugal

**Keywords:** *Physalia physalis*, collagen, collagen peptides, keratinocytes, cellular migration, inflammation

## Abstract

Collagen, the most abundant structural protein in animals, is fundamental for tissue integrity and regeneration. Conventional mammalian sources face limitations related to sustainability, safety, and ethical concerns, underscoring the need for alternative biomaterials. Marine organisms, particularly jellyfish, offer a promising eco-friendly collagen source. In this study, collagen and collagen-derived peptides were extracted from the cnidarian *Physalia physalis* and biochemically characterized. Circular dichroism demonstrated partial loss of triple-helix structure, while SDS-PAGE revealed type I collagen related α-chains together with low-molecular-weight fragments. The hydrolyzed collagen fractions exhibited keratinocyte and fibroblast cytocompatibility and increased keratinocyte migration. Moreover, *P. physalis*-derived peptides modulated inflammatory cytokine release in lipopolysaccharide-stimulated macrophages reducing tumor necrosis factor (TNF)-α by 38% and increasing interleukin (IL)-10 by 29%. Based on these results, a stable bioactive serum formulation incorporating *P. physalis* collagen peptides was developed. Overall, this work demonstrates that bioactive peptides from *P. physalis* possess immunomodulatory and regenerative potential and represent a promising new marine resource for cosmetic applications.

## 1. Introduction

Collagen is the most abundant structural protein in animals, representing up to 80% of the dry weight of human skin and forming the primary component of the extracellular matrix (ECM) in connective and interstitial tissues [1,2]. It provides tensile strength, elasticity, and structural integrity, while also regulating cell adhesion, migration, proliferation, and signaling. Due to these functions, collagen and its derivatives are widely used in the biomedical, cosmetic, and food industries [3,4].

Structurally, collagen is characterized by its unique right-handed triple helix, which is formed by three left-handed polypeptide chains, known as α-chains [5]. These chains are rich in glycine, proline, and hydroxyproline, which stabilize the helix through steric and hydrogen bonding interactions. The repetitive sequence Gly–X–Y, where X is often proline and Y is often hydroxyproline, is a hallmark of collagen molecules [6]. These ones can further self-assemble into fibrils and fibers, providing mechanical strength and resistance to tensile stress to tissues [7]. Different collagen types have been identified, each adapted to specific structural and functional requirements [8]. Among them, type I collagen is predominant in skin, tendon, bone, and ligaments, where it provides tensile strength, and forms heterotrimeric fibrils composed of two α1(I) and one α2(I) chains [9]. Type II collagen, in contrast, is a homotrimer of three α1(II) chains and is primarily found in cartilage and the vitreous humor of the eye. Type III collagen, often co-distributed with type I, contributes to the elasticity and repair capacity of tissues such as skin, blood vessels, and internal organs [10]. Other fibrillar and non-fibrillar collagens, including type IV, IX, XII, and XIV, play specialized structural and signaling roles within the extracellular matrix [8].

The structural and functional diversity of collagens is largely determined by their amino acid composition, particularly the content of proline and hydroxyproline, which stabilizes the triple-helical structure through steric constraints of the pyrrolidine ring and the formation of additional hydrogen bonds [11]. These differences in molecular architecture and stability influence fibrillogenesis, cross-linking, and thermal resistance, thereby affecting both the biological role of each collagen type and its behavior during extraction and processing [12]. This structural diversity underlies the wide-ranging biological roles of collagen and explains its versatility as a biomaterial.

Despite its importance, the use of collagen from conventional mammalian sources, including bovine and porcine, faces zoonotic risks, cultural and religious restrictions, and concerns about immunogenicity. In this context, marine organisms have emerged as sustainable alternatives for collagen production. Collagen from fish, sponges, and jellyfish exhibits lower immunogenicity and is often safer than mammalian-derived counterparts [3]. Furthermore, marine collagen valorizes ocean resources and contributes to sustainability.

Collagen has also been associated with antioxidant and regenerative activities, making it particularly attractive for the development of biomaterials, tissue regeneration scaffolds, nutritional supplements, and cosmetic formulations. As antioxidant processes represent an important mechanism to attenuate skin aging characteristics and tissue damage, collagen could be a potential source to minimize wrinkles, enhance skin elasticity, and delay visible aging signs [13].

Nevertheless, the direct application of native collagen is limited by its large molecular weight (>300 kDa), poor solubility, and limited bioavailability [14]. Hydrolyzed collagen peptides, generated through enzymatic cleavage, overcome these barriers. They have lower molecular weights (typically < 10 kDa), enabling enhanced absorption, tissue penetration, and cellular uptake [15]. In addition, collagen peptides can act not only as substrates for collagen biosynthesis but also as bioactive molecules, influencing processes such as fibroblast proliferation, keratinocyte migration, antioxidant defense [16,17,18], and modulation of inflammatory and regenerative pathways [19].

Marine organisms have become valuable biological resources in the cosmetic industry due to their richness in bioactive peptides and structural proteins that support skin regeneration and extracellular matrix remodeling [20]. Jellyfish-derived collagen has gained attention for its biocompatibility and antioxidant activity [21]. *P. physalis*, a widely distributed cnidarian species, represents an abundant and under explored source of biomolecules.

Although it resembles a jellyfish, *P. physalis* is a colonial siphonophore composed of specialized zooids responsible for feeding, locomotion, reproduction, and defense [22,23,24]. *P. physalis* can be found in tropical and subtropical waters worldwide, frequently along the coasts of North and South America, Africa, and parts of Western Europe, with occasional records in temperate regions such as the Iberian Peninsula and the Mediterranean Sea [25,26,27]. In recent years, regular sightings have been reported along the Portuguese coast and in the Azores archipelago. Ecologically, *P. physalis* is an important predator of small fish and plankton, capturing prey through cnidocytes located on its venomous tentacles [28]. However, its dysregulated proliferation due to ocean current changes and climate variability disrupts coastal ecosystems and raise public health concerns due to risk of envenomation, which pose public health concerns in marine regions [29,30]. Consequently, the extraction of collagen from *P. physalis* not only overlays the way for innovative cosmetic applications but also promotes sustainability by transforming its uncontrolled proliferation into a resourceful activity for skincare.

Despite interest in jellyfish collagen, no previous research has investigated bioactive peptides derived from *P. physalis* for cosmetic use. Therefore, the aim of this study was to evaluate the characteristics of collagen and peptides extracted from *P. physalis* and to assess their biological potential for cosmetics. Accordingly, the results were compared with commercial collagen and collagen proteins derived from other marine sources including cod skin and haddock provided by Nofima AS (Tromsø, Norway) company.

Based on the promising results of collagen peptides derived from *P. physalis*, notably their ability to promote keratinocyte migration and modulate macrophage anti-inflammatory responses, a serum bioformulation incorporating these peptides was developed in collaboration with Mesosystem (Porto, Portugal). This partnership underscores the translational potential of this research, bridging fundamental findings with industrial innovation in dermocosmetic product development.

Overall, this study identified *P. physalis* as a novel and sustainable collagen source that not only mitigates the ecological and health concerns associated with its dysregulated proliferation but also provides a valuable raw material for high-end cosmetic and regenerative applications.

## 2. Results and Discussion

### 2.1. Collagen Content

Findings demonstrated that *P. physalis* exhibits a collagen concentration of 213.04 ± 20.32 mg/g dry weight, significantly higher than that observed in haddock (98.14 ± 1.52 mg/g dry weight), cod skin (37.18 ± 14.58 mg/g dry weight), and commercially available collagen samples (35.03 ± 5.65 mg/g dry weight) (Figure 1).

Studies have reported variable collagen content in different jellyfish species. A study reported total collagen content of edible jellyfish from 122.64 to 693.92 mg/g dry weight [31]. *Rhizostoma pulmo* collagen levels ranged from 0.83 to 10.30 mg/g wet tissue [12,32]. In *Rhopilema esculentum*, extraction yields varied between 0.12% and 4.31% depending on the method employed [33,34]. In *Cyanea nozakii*, collagen represented 13.00% of the dry weight [35], while in *Catostylus mosaicus,* concentrations ranged from 4.61 to 22.47 mg/g dry weight [36]. Moreover, Nagai reported high collagen proportions from the exumbrella of *Stomolophus meleagris* (46.40%) and *Rhopilema asamushi* (35.20%) when expressed as percentage of dry lyophilized weight [37].

These results indicate that *P. physalis* represents a particularly rich and promising source of marine-derived collagen.

### 2.2. Collagen Characterization

Collagen presents a native triple-helical structure that can be observed by circular dichroism (CD) typical positive maximum absorption band at around 220 nm and minimum absorption band at 198 nm [11,38].

Intermolecular interactions stabilizing the collagen triple helix can be disrupted by dilute acids and electrostatic repulsion between similarly charged collagen monomers. The stability of collagen is further influenced by its amino acid composition, particularly the abundance of proline and hydroxyproline, which contribute to the stereochemical constraints of the pyrrolidine ring and enable the formation of additional hydrogen bonds [39]. Thus, CD spectroscopy in the far-UV region was employed to examine the impact of the extraction method on the molecular organization of collagen (Figure 2A,B). As we can observe, collagen from *P. physalis* did not exhibit a clear positive band at 220 nm, however, it displays only a negative peak at approximately 200 nm. During complete collagen denaturation, the characteristic positive peak associated with the triple helix at 220–230 nm disappears, leaving only the negative peak at 200 nm [40]. Thus, the results indicate a partial denaturation of the protein upon extraction.

Then, ATR-FTIR spectroscopy was employed as a fingerprinting technique to characterize the extracted collagen (Figure 2C).

The ATR-FTIR spectra of *P. physalis* collagen exhibited the characteristic amide bands of collagen, including amide A (~3290–3310 cm^−1^), amide B (~2925 cm^−1^), amide I (~1650 cm^−1^, C=O stretching of peptide bonds), amide II (~1535 cm^−1^, N–H bending coupled with C–N stretching), and amide III (~1240 cm^−1^, N–H deformation and C–N stretching). Additional bands were observed at 1447 cm^−1^ (CH_2_ and CH_3_ deformations), 1385 cm^−1^ (CH_3_ symmetric deformation), and 1310 cm^−1^ (CH_2_ wagging), further confirming the presence of collagen-related vibrations [41]. Compared with cod skin and haddock collagen controls, *P. physalis* displayed a noticeable reduction in amide I band intensity and a slight shift from 1652 to 1645 cm^−1^, indicating decreased hydrogen bonding organization and partial disruption of the triple-helix structure. The amide III/Amide I absorbance ratio was lower for *P. physalis* (0.62) compared with cod collagen (0.86), also suggesting increased denaturation.

These results are consistent with previous studies describing FTIR spectra of collagen from fish and other vertebrate sources. Collagen extracted from codfish skin and commercial type I collagen from rat tail and bovine skin exhibited highly similar band assignments, including amide A (3320–3329 cm^−1^, N–H stretching), amide B (~2877–2879 cm^−1^, CH_2_ asymmetrical stretching), amide I (1649–1656 cm^−1^), amide II (1546–1554 cm^−1^), and amide III at ~1239 cm^−1^ [11]. Similarly, Chen and collaborators reported acid-soluble collagen from tilapia scales and skin with amide A bands at 3318–3321 cm^−1^, amide B at 2924–2925 cm^−1^, and a triple-helix absorption ratio close to 1, confirming the presence of type I collagen [42]. In agreement with these works, the samples analyzed in this study also demonstrated the characteristic collagen absorption bands. However, some spectral differences were detected among species, particularly in cod skin collagen. These interspecies variations likely reflect differences in amino acid composition and functional groups present in the extracted material.

Then, SDS-PAGE was performed to confirm the type of collagen and its purity based on the separation of its components according to their electrophoretic mobility (Figure 2D). The molecular weight marker (lanes 1 and 6) indicates the approximate positions of the α1 and α2 and β-chains. The commercial collagen sample (lane 2) exhibited two major bands corresponding to α-chains (approximately 100–120 kDa) and a faint band near 250 kDa, corresponding to β-chains, which is typical of type I collagen [11]. The *P. physalis* collagen (lane 3) showed a distinct electrophoretic pattern with major protein bands around 120 kDa and a less intense band near 200–250 kDa, suggesting the presence of α- and β-chains similar to those observed in type I collagen. However, additional faint bands of lower molecular weight (<75 kDa) were also visible, indicating partial degradation or the presence of smaller collagen fragments. The haddock (lane 4) and cod skin (lane 5) samples displayed electrophoretic profiles characteristic of marine-derived type I collagen, with well-defined α1 and α2 chains near 100–120 kDa and β-chains close to 200–250 kDa. The cod skin collagen presented more intense α-chains compared to haddock, suggesting higher purity or a greater degree of structural integrity.

The electrophoretic profiles observed for the fish-derived collagens are consistent with those previously described for type I collagen from marine sources, which typically show two α-chains (α1 and α2) and one β-chain formed by cross-linking of α units [43]. The presence of distinct α- and β-bands in *P. physalis* collagen suggests a molecular organization similar to that of vertebrate type I collagen, although the additional lower-molecular-weight bands could reflect partial hydrolysis during extraction, as observed in jellyfish and other cnidarian collagens [32,44]. The extraction conditions used in this study, including the prolonged alkaline pre-treatment, mechanical shear during homogenization, and extended incubation with glacial acetic acid, which may promote electrostatic repulsion and acid-induced unfolding, could have contributed to structural damage of the collagen triple helix. Compared with the commercial collagen, the *P. physalis* sample exhibited a broader distribution of molecular masses, which may indicate structural heterogeneity or the coexistence of different collagen isoforms. Such variability has also been reported in other marine invertebrates, reflecting the unique biochemical composition and adaptation of collagen to the organism’s physiological environment [45].

Finally, mass spectrometry analysis was used to characterize the peptide composition of collagen samples obtained from bovine, cod, haddock, and *P. physalis*. Data were processed using the Sciex OS platform and searched with the MSFragger–Philosopher–FragPipe pipeline against curated UniProt databases corresponding to each species. Following data filtering (FDR < 1%), only peptide spectra meeting the acceptance criteria (Num Peaks ≥ 5, Intensity Sum ≥ 1 × 10^4^, and peptide mass ≥ 1000 Da) were retained for further analysis [46,47].

The obtained peptide profiles revealed a clear distinction between heavy and light fractions. The heavy fractions did not contain any collagen-related peptides and were predominantly composed of contaminant or reagent proteins such as trypsin and human keratin, which are common in proteomic workflows. In contrast, the light fractions displayed peptide sequences corresponding to collagen in the bovine, cod, and haddock samples. For *P. physalis*, no collagen peptides were detected, suggesting either the absence of collagen sequences in the reference database used (which corresponded to Physalis, Hydroidolina subclass, and not *P. physalis*) or limited sequence homology preventing confident peptide assignment. This highlights the need for improved reference datasets for marine invertebrate species.

Collagen peptides identified in the fish samples were mainly derived from type I collagen chains. Notably, the haddock sample showed peptide matches to cod collagen sequences, reflecting the shared database used for both species and their close phylogenetic relationship [48]. Comparative sequence analysis confirmed ten peptides shared between haddock and cod with complete gene correspondence, including “VVENLNVGENQIR”, “VALTLYNNEVTTEVR”, “AGETGLTGAR”, “VIDKLDVGLDNVR”, “GPAGIAGAR”, “QLGITVLGIGTR”, “LYDAGIASVFLVNREDR”, “SGVLPFSIGTR”, “GAPGALGPAGGR”, and “SSDLAQAIEYVIR” (Figure 2E).

In contrast, bovine collagen did not share identical peptide sequences with cod. However, BLAST analysis (version 2.17.0) revealed two partial sequence identities higher than 70% between bovine vs. cod (“IGQPGAVGPAGIR” vs. “GAPGALGPAGGR” and “VPQIAFVITGGK” vs. VALTLYNNEVTTEVR”) and three with 100% identity between bovine vs. haddock (“GPAGPSGPAGK” vs. “GEGGPAGPSGPAGPSGAR”, “GFSGLDGAK” vs. “GFSGLDGAK”, and “GPAGPSGPAGK vs. NGDRGEGGPAGPSGPAGPSGAR”). Interestingly, eighteen partial sequence identity higher than 70% were founded between bovine and haddock, indicating conserved motifs among vertebrate type I collagens. Between bovine and haddock, one shared peptide (GFSGLDGAK) was identified, corresponding to collagen genes COL1A1 and COL1A1b, respectively, suggesting homologous collagen isoforms across taxa.

Collectively, these peptide mapping and compositional analyses confirm the structural conservation of collagen across vertebrate sources, while also emphasizing the evolutionary and environmental factors influencing collagen composition in marine organisms.

### 2.3. Collagen Hydrolysate Production

Collagen hydrolysates have increased scientific and industrial interest due to their bioactive potential in promoting wound healing, tissue regeneration, and skin repair [49]. Their functional properties, such as antioxidant, anti-inflammatory, and cell-stimulating activities, are strongly influenced by the method of extraction and enzymatic hydrolysis used to obtain them [50,51]. Optimizing the extraction process is a crucial step in ensuring reproducible yields and consistent biological performance of collagen-derived peptides. So, next, alcalase, collagenase, and papain, commonly used proteolytic enzymes, were evaluated for their ability to hydrolyze collagen from different sources into low-molecular-weight peptides (Figure 3A). As observed, the yield of peptide extraction was approximately 2.5- to 2.7-fold higher when using collagenase (66.54 ± 0.55 mg/mL–92.58 ± 2.32 mg/mL) compared with papain (18.75 ± 0.81 mg/mL–44.04 ± 0.17 mg/mL) and alcalase (15.54 ± 0.41 mg/mL–42.74 ± 1.53 mg/mL) across all samples tested, confirming that collagenase is markedly more efficient for collagen peptide extraction. This higher yield is recognized to the enzyme’s high specificity for collagen’s triple-helical structure, enabling efficient cleavage of peptide bonds adjacent to Gly-Pro-X motifs typical of collagen fibrils [52].

These findings are in agreement with earlier studies reporting that collagenase generates a higher degree of hydrolysis and higher recovery of bioactive peptides from fish skin and other marine tissues compared to broad-spectrum proteases [53,54]. Moreover, collagenase-derived hydrolysates have been shown to possess enhanced antioxidant and wound-healing activities, likely due to their favorable peptide size distribution and amino acid composition [55].

Consequently, all subsequent experiments assessing the biological and immunomodulatory effects of collagen peptides were conducted using the collagenase-derived hydrolysates, to ensure consistent peptide profiles and optimal bio-functional potential.

The quantification of the collagen peptides extracted with collagenase was then performed (Figure 3B).

As observed, high concentrations of collagen peptides were obtained from all samples tested. The highest yield was achieved from the haddock sample (5.90 ± 0.02 mg/mL), followed by *P. physalis* (5.65 ± 0.05 mg/mL) and the commercial sample (5.58 ± 0.04 mg/mL). The cod skin extract presented a significantly lower peptide concentration (4.98 ± 0.02 mg/mL), consistent with the results observed for total collagen quantification (Figure 1).

### 2.4. In Vitro Properties of Collagen Hydrolysates

#### 2.4.1. Metabolic Activity

To evaluate the cytocompatibility and potential bioactivity of the collagen hydrolysates obtained, metabolic activity assays were performed on HaCaT human keratinocytes, BJ-t5a human fibroblasts, and L929 mouse fibroblasts. The cells were exposed to increasing concentrations (0.01, 0.05, 0.10, and 0.50 mg/mL) of collagen hydrolysates obtained from the commercial, *P. physalis*, cod skin, and haddock samples (Figure 4).

In HaCaT cells, a tendency of concentration-dependent effect was observed (Figure 4A). At lower concentrations (0.01–0.10 mg/mL), no significant cytotoxicity was detected across any of the hydrolysates, indicating good biocompatibility. At the highest concentration (0.50 mg/mL), metabolic activity increased in cells treated with the commercial hydrolysate, followed by the *P. physalis* and cod skin peptides, suggesting a mild stimulatory effect on keratinocyte metabolism. These results indicate that the tested hydrolysates do not compromise cell viability and may promote keratinocyte activation at higher doses. The same pattern was observed in BJ-t5a fibroblasts (Figure 4B), where commercial and *P. physalis* hydrolysates induced the most pronounced metabolic activity increase, reaching up to ~150%. These findings suggest a strong stimulatory effect on fibroblast metabolism, consistent with the role of collagen peptides in stimulating fibroblast activity and extracellular matrix synthesis. For L929 fibroblasts, similar to HaCaT cells, the hydrolysates exhibited no cytotoxic effects at any concentration.

Overall, all tested collagen hydrolysates demonstrated positive cytocompatibility and potential metabolic stimulatory effects, particularly at higher concentrations. Based on these findings, the concentration of 0.50 mg/mL was selected for subsequent in vitro experiments to further investigate the collagen biopeptides’ biological activities.

#### 2.4.2. Cell Migration

Keratinocyte migration is a fundamental biological process in skin regeneration and wound healing [56]. Following injury, keratinocytes at the wound margins proliferate and migrate to cover the exposed dermis, initiating re-epithelialization and restoring the skin’s barrier function [57]. Impaired keratinocyte migration is a hallmark of chronic wounds, which fail to progress beyond the inflammatory phase of healing and are often associated with burns, infections, or metabolic and genetic disorders [58,59]. Beyond wound repair, cell migration also underlies essential physiological mechanisms, such as immune cell trafficking and tissue homeostasis [60].

Marine-derived collagen peptides have emerged as promising bioactive materials for promoting cell migration, accelerating tissue regeneration and healing by facilitating cell adhesion and motility [61,62]. Therefore, the migratory ability of keratinocytes (HaCaT cell line) in the presence of different collagen peptides was evaluated using a scratch wound-healing assay. Based on the effects of collagen peptide concentration on cell metabolic activity, cells were treated with 0.5 mg/mL of each collagen peptide, and wound closure was monitored at 0 h, 6 h, and 24 h (Figure 5).

At the initial time point (0 h), a clear wound gap was observed in all groups. Progressive closure of the scratch was evident over time, with cell migration becoming prominent after 6 h and nearly complete by 24 h in most treatments (Figure 5A). At 6 h, the control group exhibited minimal wound closure (5.17 ± 1.72%), and the commercial peptide group showed a similar low closure (3.89 ± 1.68%). Cells treated with *P. physalis*-derived peptides showed a comparable closure to the control (5.14 ± 0.96%), whereas cod skin- and haddock-derived peptides markedly enhanced migration, with wound closure of 13.74 ± 0.47% and 18.93 ± 0.82%, respectively. By 24 h, wound closure had progressed substantially in all groups. However, all the experimental groups presented significantly higher wound closure compared to the control group (80.46 ± 1.70%). Treatments with commercial, *P. physalis*, cod skin, and haddock peptides achieved nearly complete closure, with values of 85.11 ± 1.58%, 93.08 ± 0.63%, 98.10 ± 1.90%, and 95.06 ± 0.82%, respectively (Figure 5B). Cod skin peptides demonstrated the highest efficacy, promoting almost complete wound closure within 24 h.

The increased wound closure observed aligns with previous findings demonstrating the beneficial effects of marine-derived collagen on wound healing and tissue regeneration [63]. Marine collagens, particularly those obtained from fish skin, possess low molecular weight peptides that exhibit high bioavailability and biological activity, enhancing cell adhesion, proliferation, and migration [64,65]. Hu and collaborators [64] reported that marine collagen peptides at 50 μg/mL promoted wound closure in vitro within 12 h, an effect comparable to epidermal growth factor (EGF) at 10 ng/mL, a well-established promoter of epithelial regeneration [62]. Similarly, in vivo studies using tilapia-skin-derived collagen peptides showed accelerated wound healing in rats and rabbits compared with untreated controls [62,66].

Overall, these findings reinforce the potential of bioactive motifs present in collagen peptides from *P. physalis*, haddock, and cod skin to promote keratinocyte motility and accelerate re-epithelialization processes.

#### 2.4.3. Macrophage-Derived Inflammatory Cytokines

Macrophages are central modulators of inflammation and tissue repair, secreting cytokines that determine the balance between pro-inflammatory, including IL-1β, TNF-α, and IL-6 and pro-regenerative responses, including IL-10 and TGF-β1 [67]. Assessing how collagen hydrolysates influence macrophage cytokine production is therefore essential to predict their immunomodulatory profile and safety for wound-healing or dermocosmetic applications. Cytokine release was quantified in resting (non-stimulated) (Figure 6) and lipopolysaccharide (LPS)-stimulated macrophages (Figure 7) exposed to different collagen biopeptides to evaluate potential pro- or anti-inflammatory effects.

As expected, stimulation with 1 µg/mL LPS significantly increased secretion of canonical pro-inflammatory cytokines compared with the non-stimulated control (Figure 6A–D). In naïve macrophages, collagen peptides from the commercial hydrolysate induced an increased production of pro-inflammatory cytokines, showing a similar pattern to the LPS-positive control (Figure 6A–C). All peptide samples promoted some degree of TNF-α release (Figure 6C).

The increase in pro-inflammatory cytokines observed in naïve macrophages exposed to certain collagen peptide samples, particularly the commercial hydrolysates, may be explained by two complementary mechanisms. First, macrophages exhibit high sensitivity to collagen fragments, which can act as danger-associated molecular patterns (DAMPs) and activate pattern-recognition receptors such as toll-like receptors 2 and 4 (TLR2 and TLR4). This mechanism, described for collagen-derived peptides, can trigger low-grade cytokine release even in the absence of LPS stimulation [68]. Second, differences in molecular-weight distribution and the potential presence of residual processing components in commercial hydrolysates may contribute to stronger basal macrophage activation compared to more homogeneous marine-derived samples. Although direct comparative evidence is limited, variability in hydrolysate composition is known to influence biological responses. Notably, this mild immune activation was not observed with *P. physalis*, cod skin, or haddock-derived peptides, suggesting that marine collagen hydrolysates may offer a more favorable safety and immunomodulatory profile.

While LPS alone triggered high IL-1β and TNF-α secretion, treatment with collagen peptides attenuated these responses. Peptides derived from cod skin and haddock elicited the strongest reductions in IL-1β compared with LPS alone (Figure 6A). In contrast, IL-6 levels remained elevated following LPS stimulation and were not consistently modulated by peptide treatment (Figure 6B), suggesting this cytokine is less responsive to peptide-mediated regulation under these conditions. Peptides from commercial and haddock sources maintained or slightly enhanced TGF-β1 production compared with LPS alone (Figure 6D). Moreover, several peptide treatments, particularly those from *P. physalis* and cod skin, tended to increase IL-10 secretion relative to the LPS control, indicating a possible shift toward an anti-inflammatory or pro-regenerative phenotype (Figure 6E).

In LPS- LPS-stimulated macrophages (Figure 7A–E), only LPS and the commercial collagen peptides increased production of the pro-inflammatory cytokines IL-1β and TNF-α compared to the control (Figure 7A,B). Conversely, collagen peptides from *P. physalis*, cod skin, and haddock did not significantly affect the production of IL-1β, IL-6, TNF-α, or IL-10, suggesting that these marine-derived peptides do not promote inflammation in activated macrophages. The high variability observed in IL-10 levels may reflect inherent biological heterogeneity combined with the small sample size, as well as technical variability associated with macrophage passage differences, batch-dependent LPS responsiveness, and ELISA sensitivity.

This immunomodulatory profile aligns with previous studies demonstrating anti-inflammatory and wound-healing-promoting effects of marine collagen peptides [3]. Both in vitro and in vivo studies have shown that low-molecular-weight marine collagen peptides can reduce pro-inflammatory TNF-α and IL-1β expression and accelerate tissue repair through upregulation of TGF-β signaling [69,70,71]. Similarly, collagen-derived peptides have been reported to increase TGF-β1 and other regenerative factors that promote extracellular matrix deposition and re-epithelialization [72,73].

Overall, the data indicate that collagen biopeptides, particularly those derived from *P. physalis*, cod skin and haddock, are non-pro-inflammatory under basal conditions and can modulate macrophage responses under inflammatory stimulation. These peptides most effectively reduced LPS-induced IL-1β and TNF-α while maintaining or slightly enhancing TGF-β1 and IL-10 levels, suggesting a shift toward a less inflammatory and more regenerative macrophage phenotype. Collectively, these findings support the potential of selected marine collagen hydrolysates as immunomodulatory ingredients for wound-healing cosmetic formulations.

### 2.5. Formulation of a Bioactive Serum Enriched with Collagen Hydrolysates

Given the promising bioactivity of collagen peptides observed in in vitro assays, the industrial partner Mesosystem S.A. developed two prototype formulations: a serum containing collagen biopeptides and a water-in-oil nanoemulsion serum encapsulating collagen biopeptides. These prototypes were designed to evaluate the stability and applicability of collagen hydrolysates in cosmetic formulations.

For the collagen biopeptide serum, formulation optimization ensured compatibility within pH 5.8 ± 0.2, the expected pH range for dermocosmetic products. The serum matrix was structured using a polymeric base, temperature between 22–55 °C, viscosity of 2800 ± 240 cP, and no phase separation during 30-day accelerated storage (4–45 °C cycling), providing stability and desirable rheological properties (Appendix A). As delineated in Appendix A, the formulation process involved dissolvation of collagen biopeptides in the aqueous phase, followed by incorporation into the polymeric matrix, resulting in a stable, homogenous, and bioactive serum formulation. This stability is attributed to the optimized polymeric network, which provides adequate structuring to entrap the aqueous phase and evenly disperse the collagen biopeptides. The selected polymer creates strong intermolecular interactions that prevent water migration and maintain a stable gel-like matrix, ensuring homogeneity throughout the storage period.

A second serum prototype was developed using a water-in-oil (W/O) nanoemulsion, designed to encapsulate collagen biopeptides within an oily continuous phase (Appendix A). The formulation required careful selection of surfactants and oils to achieve the ideal hydrophilic–lipophilic balance, targeted at values ≤ 9 to favor stable W/O nanoemulsion formation [74]. This approach enhances the protection and controlled release of collagen peptides, potentially improving skin penetration and bioavailability [75]. The stability assessment included visual inspection and storage testing, with both indicating no phase separation. In this prototype, phase stability results from the correct ratio of surfactants, which provides sufficient interfacial tension reduction, and from the formation of a tight oil-continuous phase that restricts droplet coalescence or creaming, thereby maintaining the structural integrity of the nanoemulsion.

Comprehensive characterizations of these formulations, including stability testing, physicochemical analysis, and in vitro assays, are being carried out by Mesosystem S.A. This collaboration within an academic–industrial consortium represents a significant step toward the translation of fundamental research into practical applications within the cosmetic biotechnology sectors.

Collagen denaturation can affect the mechanical strength and firmness of biomaterials, as intact fibrils can assemble into load-bearing networks. However, denatured collagen and low-molecular-weight hydrolyzed peptides retain significant bioactivity, as they are more readily absorbed by skin cells and can act as signaling molecules to stimulate fibroblast proliferation, modulate cytokine release, and promote extracellular matrix remodeling. In this study, the hydrolyzed collagen fractions demonstrated the ability to enhance cell migration and regulate macrophage cytokine production, supporting their potential role in cosmetic applications. Therefore, while denaturation may reduce the contribution to mechanical firmness, it does not compromise the bioactive and functional properties of the collagen for topical cosmetic use.

The integration of marine-derived collagen peptides into cosmetic formulations highlights the potential of these natural biomolecules to bridge scientific innovation and industrial development, contributing to the advancement of sustainable and bioactive skincare products.

## 3. Materials and Methods

### 3.1. Sampling

*P. physalis* was collected in the Azores coastal area. Following collection, the samples were stored at −80 °C and protected from light for further analysis. Collagen from haddock (*Melanogrammus aeglefinus*) and cod (*Gadus morhua*) provided by the Nofima company, along with a commercial collagen reference sample (Advanced BioMatrix, 5162-1GM, Lot No. 8152, Carlsbad, CA, USA), were used for comparison. Collagen extracted from cod and haddock skin was used, as these species are widely used commercial marine collagen sources with well-described biochemical characteristics. Their inclusion provides a relevant performance baseline to evaluate the novelty and functional quality of *P. physalis* collagen.

### 3.2. Collagen Extraction

*P. physalis* was first washed with 0.1 M sodium hydroxide (NaOH) (1:10, *w*/*v*) under stirring for 24 h. The NaOH solution was then removed and discarded, and the samples were rinsed thoroughly with cold distilled water (Milli-Q Water Purification System, Darmstadt, Germany) until the pH stabilized at approximately 7.0. The tissues were subsequently homogenized using a food processor (Yämmi Robot, San Francisco, CA, USA), and the homogenate was solubilized in 0.5 M glacial acetic acid (695092, Sigma-Aldrich, St. Louis, MO, USA, 99.8% purity). Specifically, 100 g of homogenized *P. physalis* tissue was mixed with 500 mL of distilled water and 2.5 mL of 0.5 M glacial acetic acid, followed by continuous stirring at 10 °C for at least 24 h. The resulting solution was centrifuged at 7500 rpm for 45 min at 4 °C to separate the supernatant from the pellet. The supernatant was neutralized to a pH of approximately 6.0 using concentrated NaOH (43617, Sigma-Aldrich), and collagen was precipitated with 2 M sodium chloride (NaCl) (S9888, Sigma-Aldrich). The mixture was incubated for 1 h and centrifuged again under the same conditions. The pellet was collected, washed twice with NaCl, and finally rinsed with distilled water. The pellet was then dissolved in 0.5 M acetic acid, supplemented with 7 mg pepsin (107192, Sigma-Aldrich), and stirred for 24 h. The pellet was then solubilized with 0.5 M acetic acid three times, followed by a second precipitation step with 0.9 M NaCl on ice. After resting for 1 h, the precipitate was centrifuged, washed twice with NaCl, and a gradual dialysis with 0.1 M, 0.05 M, and 0.025 M acetic acid was performed. The extracted collagen was subsequently lyophilized and stored under appropriate conditions until further use. The fish skin collagen was extracted as acid-soluble collagen, with a method similar to that of Matmaroh and co-authors [76]. Briefly, the skins were washed with water and treated with NaOH, before neutralization with dH_2_O and extraction with acetic acid. The extracted collagen was obtained by centrifugation and precipitated using NaCl. The precipitate was dialyzed against dH_2_O and lyophilized. All the steps were performed at 10 °C to avoid collagen degradation.

### 3.3. Collagen Quantification

Collagen concentration was quantified using a hydroxyproline colorimetric assay with L-hydroxyproline standards (0–100 µg/mL, Sigma-Aldrich). Tissue collagen content was calculated using a conversion factor of 7.46 based on the average hydroxyproline composition of collagen.

### 3.4. Collagen Hydrolysate Preparation

The extracted collagen (60 mg) was resuspended in 10 mL of phosphate-buffered saline (PBS 1×, pH 7.4; PAN Biotech, Aidenbach, Germany). Commercial collagenase 1:6 (*w*/*w*) (LS0004180, PAN Biotech), papain 1:6 (*w*/*w*) (P4762, Sigma-Aldrich), and alcalase 1:6 (*w*/*w*) enzyme-to-substrate ratios (126741, Sigma-Aldrich) (10 mg; final enzyme concentration 1 mg/mL) were added to test the most efficient enzyme. The mixture was incubated for 24 h at 37 °C under agitation (100 rpm) in a water bath. The reaction was terminated by heating the solution to 100 °C for 10 min. The hydrolysates were cooled to room temperature and collected by centrifugation at 5000 rpm for 10 min. The resulting peptides were quantified using a NanoDrop spectrophotometer (Thermo Fisher Scientific, Waltham, MA, USA).

### 3.5. Collagen Characterization

The structural characterization of the extracted material was performed using Attenuated Total Reflection–Fourier Transform Infrared Spectroscopy (ALPHA II-Bruker compact FTIR spectrometer, Ettlingen, Germany) and Circular Dichroism (CD) spectroscopy. ATR-FTIR spectra were recorded with a resolution of 1 cm^−1^ in the range of 4000–400 cm^−1^, averaging 64 scans per measurement. For CD analysis, the extracted material was dissolved in acetic acid, and ellipticity values were corrected against the acetic acid baseline. CD spectra were recorded at 180 to 280 nm on a Jasco J-1500 dichrograph (Jasco Corp., Tokyo, Japan) using a 0.1 cm path–length cuvette.

#### 3.5.1. Protein Characterization and Peptide Mapping

Data were acquired and analyzed with the Sciex OS version 3.1.6 software (Sciex), using the LCMS Peptide Reconstruct (with peak finding) from the BioTool kit. Then, the following (cumulative) acceptance criteria were applied: Num Peaks ≥ 5; Int Sum ≥ 1 × 10^4^; mass ≥ 1000. Generated mass spectra (.wiff2 files) were converted to mzML format using the msconvert.exe tool from ProteoWizard (v. 3.0.23111-67c7064). Conversion was performed using vendor default parameters for peak picking at the MS1 level. All searches were done considering the following settings: MSFragger’s built-in mass calibration; N-terminal methionine clipping, deisotoping, and default neutral loss removal; only y- and b- ions were considered; tryptic peptides only, up to 2 missed cleavages, with cysteine carbamidomethylating specified as a fixed modification and methionine oxidation and N-terminal acetylation as variable modifications; and allowing a maximum of 3 variable modifications per peptide; require at least 4 matched fragment ions for a PSM to be reported. All searches were performed using MSFragger (v.4.1), Philosopher (v.5.1.1), and FragPipe (v.22.0). The protein database was created from the protein sequences of UniProt (downloaded August 2024) from Bos taurus (TID: 9913, Ref. Proteome UP000009136, 59,265 entries), Haddock and Cod (TID: 8049, Ref. Proteome UP000694546, 61,777 entries), or *Physalis* (TID: 37516, Hydroidolina subclass, 54905 entries), each appended with their respective reversed protein sequences as decoys, and common contaminants (cRAP proteins sequences from gpmDB and contaminants from MaxQuant, https://www.maxquant.org/). Bos taurus collagen was included as a commercial reference standard commonly used in cosmetics. Protein identification was considered with FDR < 1%.

#### 3.5.2. Bioinformatics Analysis

The peptides identified as collagen from the Mass Spectrometry (X500B positive Q-TOF MS, twin-spray ion source (Sciex, Framingham, MA, USA)) study were selected from the total pool of identified sequences. Protein sequence analyses were conducted in Python (version 3.8) using the Biopython library (version 1.7). Searches were performed with BLASTp to execute local searches against the selected protein sequence database. Collagen peptides were compared between Haddock, Bos Taurus and *P. physalis* entries.

### 3.6. Sodium Dodecyl Sulfate Polyacrylamide Gel Electrophoresis

Sodium dodecyl sulfate polyacrylamide gel electrophoresis (SDS-PAGE) was performed to evaluate the protein profile of the extracted collagen. Briefly, 20 µL of each sample was mixed with 5 µL of loading buffer (Nzytech, Lisbon, Portugal) and denatured at 95 °C for 5 min. Collagens were resolved on 12% polyacrylamide gels at 110 V for 60 min using a Mini-PROTEAN Tetra Cell electrophoresis system (catalog no. 165-8030, Bio-Rad, Hercules, CA, USA). After electrophoresis, the gels were developed using a silver staining protocol to visualize the protein bands. Molecular weight estimation was carried out by comparing the bands with standard protein markers.

### 3.7. Macrophage Activation

#### 3.7.1. Cell Maintenance

The macrophage cell line J774A.1 was used to assess cell viability and cytokine secretion following incubation with the extracts. The cell line was routinely cultured in DMEM high glucose, supplemented with 10% heat-inactivated FBS, 2 mM glutamine, 1 mM sodium pyruvate, and 25 mM HEPES buffer. The cultures were maintained in tissue culture flasks (Nagle Nunc, Int., Hereford, UK) with a humidified atmosphere containing 5% CO_2_ at 37 °C (Binder CB150; Tuttlingen, Germany).

#### 3.7.2. Cell Viability

After confluent growth, macrophage cells were washed with fresh medium and recovered by scraping. Viable cells were counted by Trypan blue exclusion in the hemocytometer and resuspended in DMEM to a final concentration of 1 × 10^6^ cells/mL. A volume of 300 µL of the macrophage suspension was then cultured in 48-well tissue culture plates. Cells were incubated overnight with or without lipopolysaccharides (LPS, Sigma) at a final concentration of 1 µg/mL. Following incubation, supernatants were collected, and the cells were treated with the extracts. After 24 h of treatment, the supernatants were stored at −20 °C for cytokine quantification. The metabolic viability of the cells was determined using the MTT assay, as described by Madesh et al. [77]. Formazan crystals formed were dissolved in a DMSO: ethanol (1:1) solution, and absorbance was measured at 570 nm.

#### 3.7.3. Cytokine Quantification

The concentrations of tumor necrosis factor (TNF)-α (# 88-7324-88, Invitrogen, Carlsbad, CA, USA), interleukin (IL)-1β (# 88-7013A-88, Invitrogen), IL-6 (# 88-7064-88, Invitrogen), IL-10 (# 88-7105-88, Invitrogen) and transforming growth factor (TGF)-β1 (# 88-8350-88, Invitrogen) in cell culture supernatants were quantified using the corresponding Mouse Uncoated ELISA kit, following the manufacturer’s instructions. The absorbance was measured at 450 nm and 570 nm at room temperature on an Infinite M200 NanoQuant microplate reader (Tecan, Männedorf, Switzerland, EUA). Controls included cells incubated only with DMEM and with LPS at a final concentration of 1 µg/mL (positive control).

### 3.8. In Vitro Assays

#### 3.8.1. Cell Maintenance

Non-cancerous cell lines, including mouse fibroblast (L929, ATCC^®^ CCL-1), human immortalized keratinocytes (HaCaT) and human hTERT immortalized fibroblast (BJ-5ta, ATCC^®^ CRL-4001), were used to evaluate distinct biological functions. Cells were maintained in DMEM medium supplemented with 10% FBS and 1% penicillin/streptomycin (PS, Sigma). All cell lines were cultured and grown to ~80% confluence and were then sub-cultured and maintained at 37 °C with 5% CO_2_ in a humidified environment.

#### 3.8.2. Cell Metabolic Assay

Upon confluence, cells were trypsinized (PAN Biotech) and seeded into 96-well plates at 4 × 10^4^ cells/mL cellular density. The cell lines in this study were incubated with supplemented DMEM and collagen samples with different concentrations of collagen (0.01, 0.05, 0.10, and 0.50 mg/mL) for a period of 24 h. The metabolic activity was measured using the resazurin reduction assay. The supernatant was replaced with 200 µL of culture media containing resazurin (0.5 mM in PBS). After 2 h of incubation at 37 °C, the fluorescence product, resorufin, was measured using a spectrometric microplate reader (Synergy, HT BioTek Instruments, Inc., Winooski, VT, USA) at 560 nm excitation and 590 nm emission wavelength. Each experiment was performed in triplicate. The percentage of cell viability metabolic activity was calculated by relating the sample values to untreated controls (PBS).

#### 3.8.3. Cell Migration

The cell lines were seeded into a 24-well plate and incubated at 37 °C in 5% CO_2_ atmosphere until confluence was reached. A scratch on the confluent cell layer was performed using a 10 μL pipette tip. The culture medium was then replaced with the different collagens dissolved in culture medium at a concentration of 0.5 mg/mL. Cell migration was monitored microscopically, and images were acquired at 0 h, 6 h and 24 h, in the same region. The area of the scratch was measured at different time points using Image J’s plugin, MRI Wound Healing Tool. The cellular migration rate was calculated according to the following equation:$$Wound\;closure\;\left(\%\right)\;=\;\left(\frac{At\left(0\;h\right)\;-\;At\left(\Delta h\right)}{At\left(0\;h\right)}\right)\;\times \;100$$
where At (0 h) = is the area of the wound measured immediately after scratching (time zero), and At (∆h) is the area of the wound measured at h hours after the scratch is performed.

### 3.9. Statistical Analysis

Statistical comparisons were analyzed in SPSS Statistics, version 27.0.1.0 (IBM SPSS Statistics Software, Chicago, IL, USA), and GraphPad Prism, version 8.4.0 (GraphPad Software, San Diego, CA, USA). Normality was measured by the Shapiro-Wilk statistical test and was assumed when *p* > 0.05. All data were evaluated by a one-way analysis of variance (ANOVA). When statistical differences were identified, the variables were compared using Tukey’s multiple range test. As most datasets did not meet normal distribution, were performed using the Wilcoxon test with Bonferroni correction. When homogeneity of variances was not applied, Welch’s test, followed by a post hoc Games–Howell, test was performed. Results are presented as the mean ± standard deviation (SD) and the level of statistical significance was considered at *p* < 0.05.

## 4. Conclusions

This study demonstrated *P. physalis* as a novel and sustainable source of marine collagen with distinct structural and biological properties for cosmetic applications. The extracted collagen and derived peptides exhibited high yield and biocompatibility, promoting keratinocyte migration, fibroblast activity, and macrophage-mediated anti-inflammatory responses. The successful development of a bioactive serum formulation in collaboration with Mesosystem (Porto, Portugal) highlights the translational potential of this marine-derived biomaterial for dermocosmetic applications. Future research should optimize extraction parameters to minimize structural damage and improve the preservation of native collagen architecture, as well as explore the molecular mechanisms underlying the bioactivity of *P. physalis* peptides, optimize large-scale extraction and purification methods, and evaluate their in vivo efficacy and safety in skin regeneration and wound-healing models. Developing these investigations could consolidate *P. physalis* as a sustainable and high-value collagen source within the blue biotechnology sector.

## Figures and Tables

**Figure 1 ijms-27-00033-f001:**
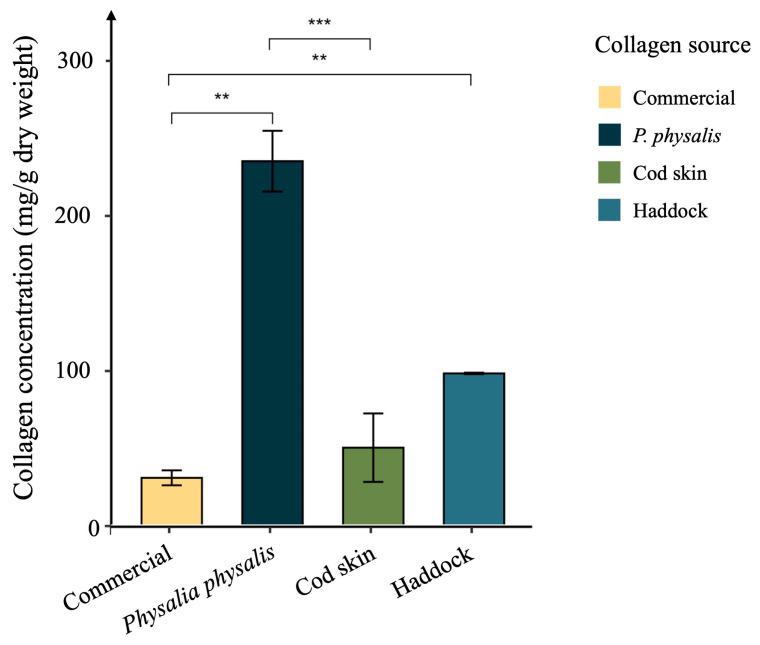
Collagen concentration obtained from *P. physalis* and comparison with described high sources of collagen, including a commercial sample, cod fish, and haddock. Data are presented as mean ± standard error from three independent experiments. Differences between each collagen source were considered at *p* < 0.05, according to the analysis of variance (ANOVA) followed by a parametric test. ** *p*-value < 0.01, *** *p*-value < 0.001.

**Figure 2 ijms-27-00033-f002:**
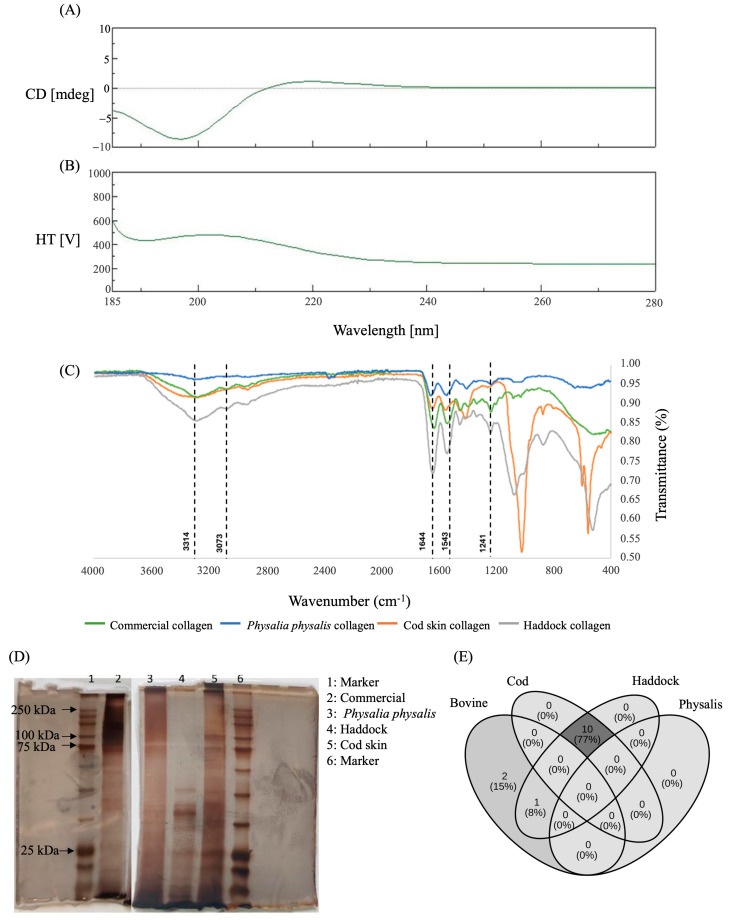
Characterization of collagen. Circular dichroism spectra representing (**A**) a triple-helix structure in collagen with a positive band maximum ellipticity at 220 nm and (**B**) the spectra from the collagen extract from *P. physalis*. (**C**) Attenuated total reflection–Fourier transform infrared spectroscopy of collagens from commercial, *P. physalis*, cod skin, and haddock samples with exhibition of the main vibrations of collagen molecular organization, amide A, amide B, amide I, amide II and amide III. (**D**) Sodium dodecyl sulfate-polyacrylamide electrophoresis of collagens from commercial, *P. physalis*, cod skin, and haddock samples evidences the presence of monomer ((α1)2(α2)) and dimers (β) typical of collagen type I. (**E**) Peptide mapping and comparative sequence alignment of collagens from bovine, cod, haddock, and *P. physalis*. The Venn diagram represents shared and unique peptides identified across species. Overlapping regions correspond to conserved collagen α-chain peptides, highlighting the sequence similarity between cod and haddock, and partial homology with bovine collagen. No collagen peptides were detected in *P. physalis*, likely due to the absence of specific entries in the reference database.

**Figure 3 ijms-27-00033-f003:**
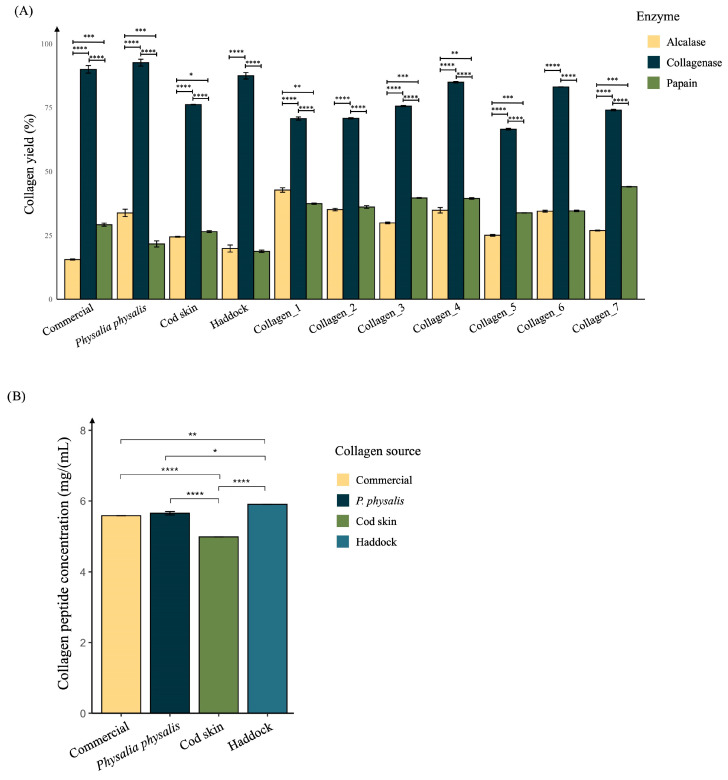
Production of collagen hydrolysates. (**A**) Yield of collagen peptide extraction using alcalase, collagenase, and papain proteolytic enzymes. Differences between each enzyme within the collagen source were considered *p* < 0.05. (**B**) Collagen peptide concentration obtained from *P. physalis* and comparison with described high sources of collagen, including a commercial sample, cod skin, and haddock. Differences between each collagen peptide source were considered at *p* < 0.05. Data are presented as the mean ± standard error from three independent experiments. Analysis of variance (ANOVA) followed by a parametric test was performed. * *p*-value < 0.05, ** *p*-value < 0.01, *** *p*-value < 0.001, **** *p*-value < 0.0001.

**Figure 4 ijms-27-00033-f004:**
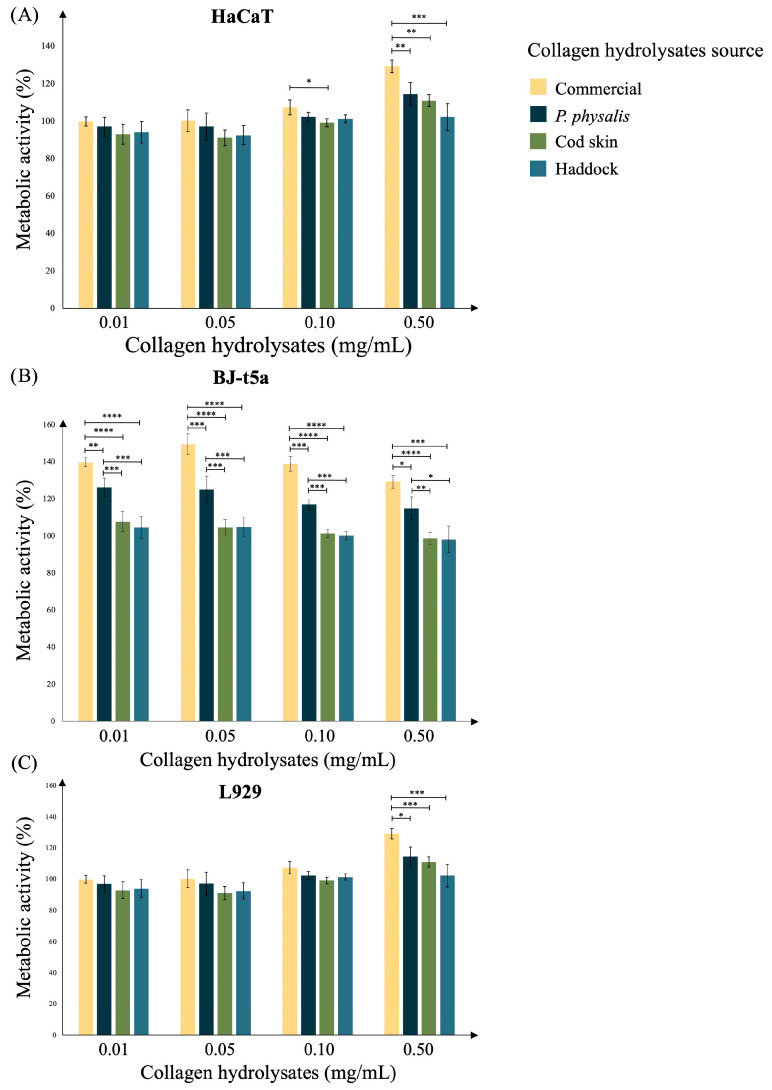
Effect of collagen peptides from commercial, *P. physalis*, cod skin, and haddock samples on the metabolic activity of HaCaT (**A**), BJ-t5a (**B**), and L929 (**C**) cells measured through the resazurin reduction into fluorescent resorufin assay by metabolically active cells after 24 h of exposure to different concentrations including 0.01, 0.05, 0.10, and 0.50 mg/mL. Data are presented as the mean ± standard error from five independent experiments. Differences within each collagen hydrolysate concentration between each collagen hydrolysate source were considered at *p* < 0.05, according to the analysis of variance (ANOVA) followed by a parametric test. * *p*-value < 0.05, ** *p*-value < 0.01, *** *p*-value < 0.001, **** *p*-value < 0.0001.

**Figure 5 ijms-27-00033-f005:**
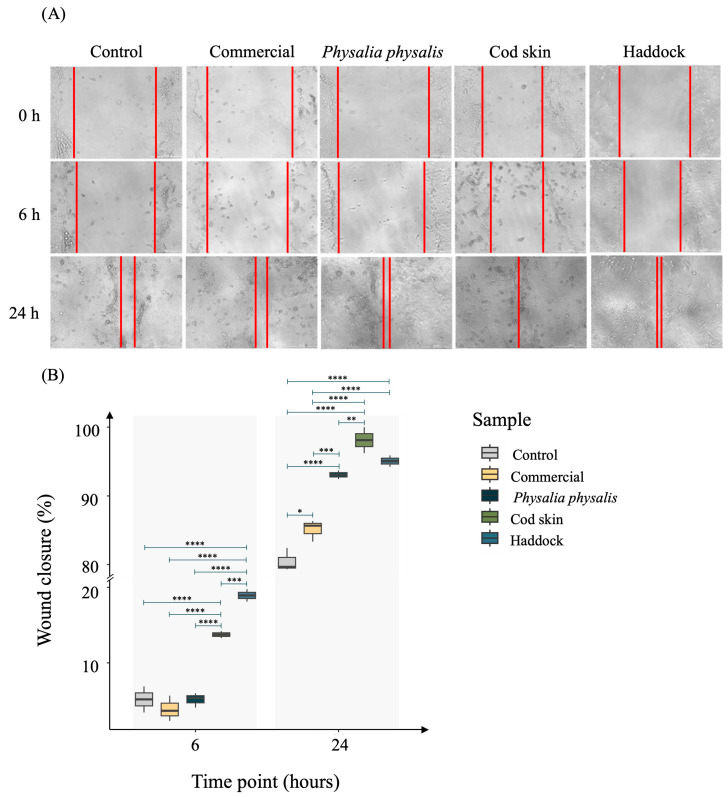
Effect of 0.50 mg/mL collagen peptides from commercial, *P. physalis*, cod skin, and haddock samples on scratch closure. (**A**) Representative scratch wound assay images, in which red lines indicate the boundaries of the cell migration region and (**B**) quantification of wound closure of HaCaT cells after treatment with collagen peptides from a commercial sample, *P. physalis*, cod skin, and haddock after 6 h and 24 h. Control cells refer to those treated with culture medium only, without peptide supplementation. Wound closure percentage was expressed as mean ± standard error from three independent experiments. Comparisons between treatments and respective controls at each time point were performed using the Wilcoxon test with Bonferroni correction and considered at *p* < 0.05. * *p*-value < 0.05, ** *p*-value < 0.01, *** *p*-value < 0.001, **** *p*-value < 0.0001.

**Figure 6 ijms-27-00033-f006:**
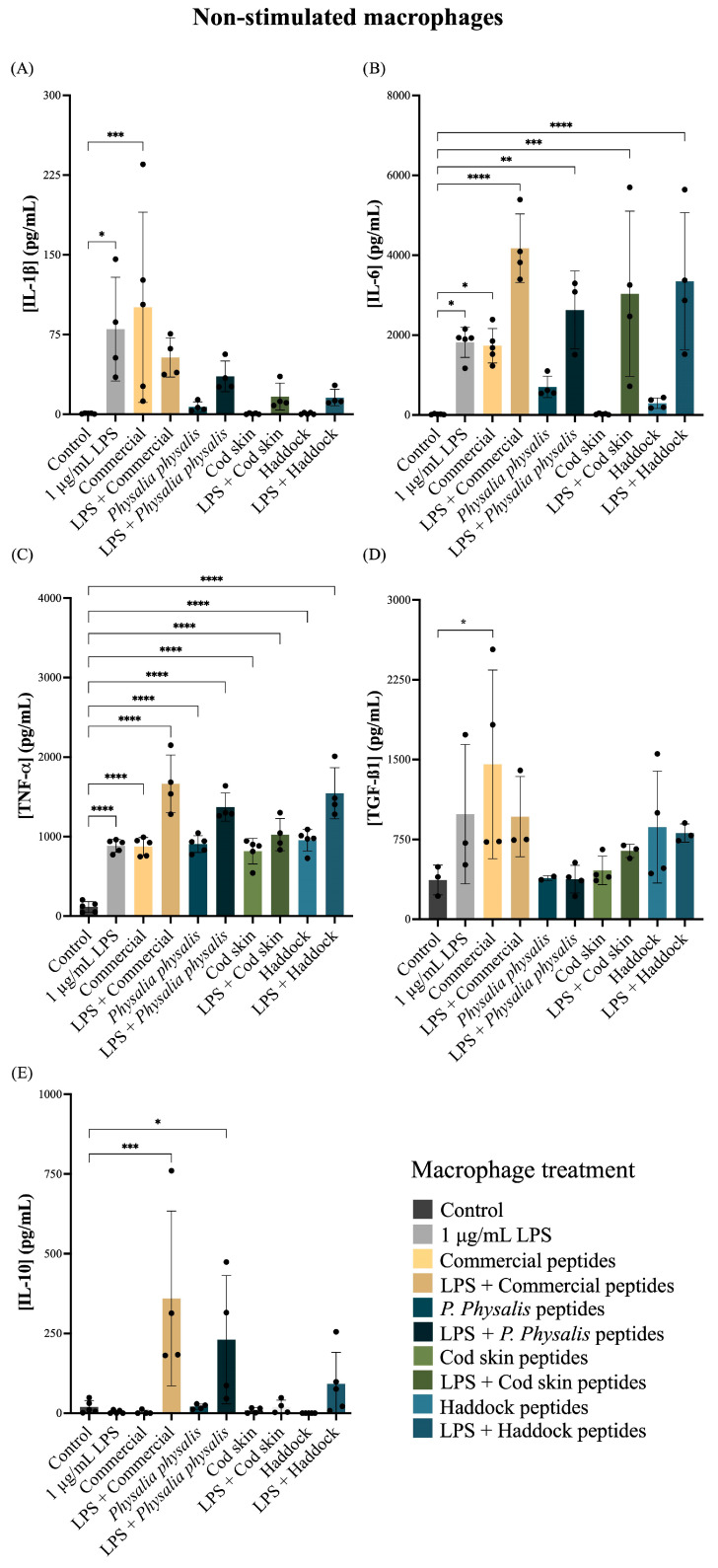
Effect of 0.50 mg/mL collagen peptides from commercial, *P. physalis*, cod skin, and haddock samples on cytokine production by naïve macrophages, including IL-1β (**A**), IL-6 (**B**), TNF-α (**C**), TGF-β (**D**), and IL-10 (**E**). Untreated medium (Control) was used as a negative control and 1 µg/mL LPS was used as a positive control. Data are presented as mean ± standard deviation from five independent experiments. Differences between macrophages treated with different conditions were considered at *p* < 0.05, according to the analysis of variance (ANOVA) followed by a non-parametric test. * *p*-value < 0.05, ** *p*-value < 0.01, *** *p*-value < 0.001, **** *p*-value < 0.0001. IL: interleukin, LPS: lipopolysaccharide, TGF: transforming growth factor, TNF: tumor necrosis factor.

**Figure 7 ijms-27-00033-f007:**
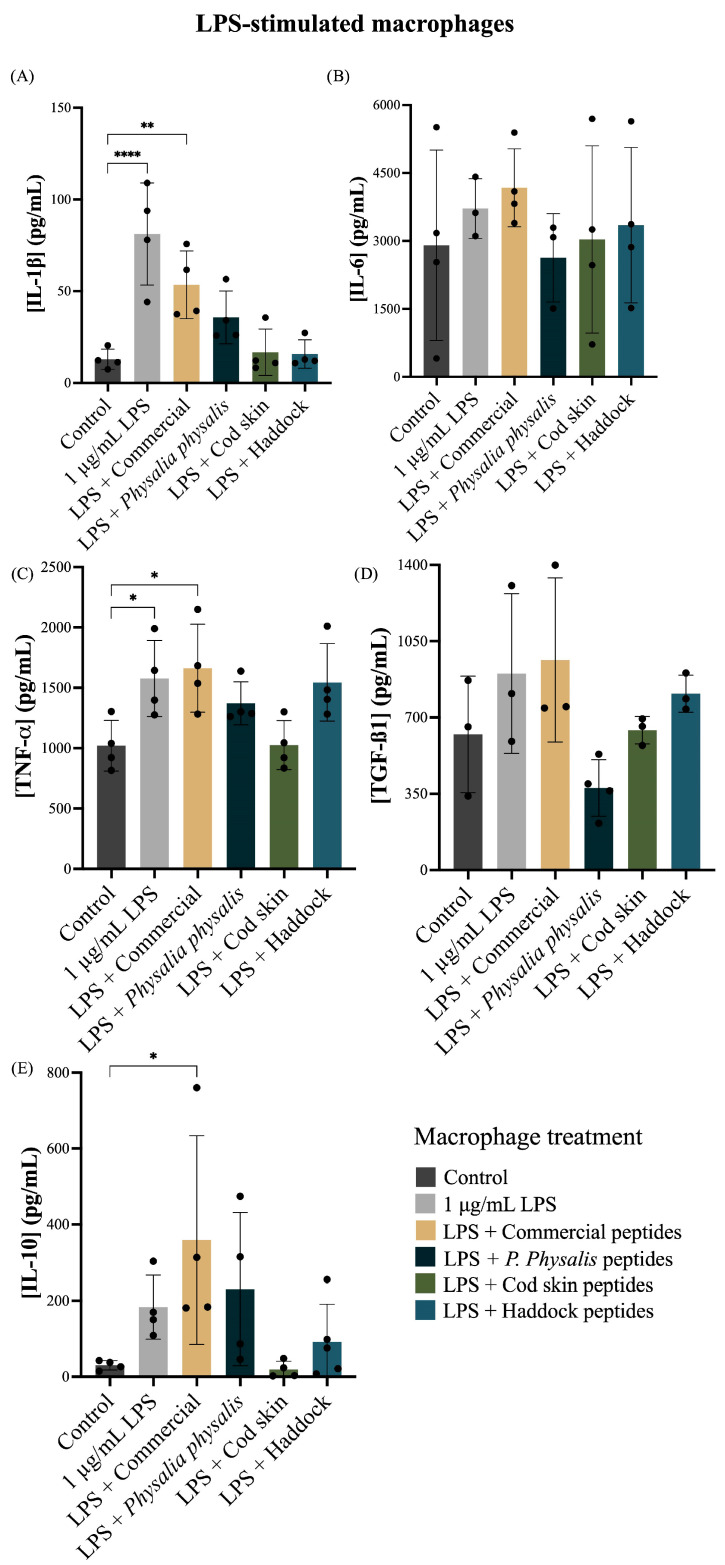
Effect of 0.50 mg/mL collagen peptides from commercial, *P. physalis*, cod skin, and haddock samples on cytokine production by LPS-stimulated macrophages, including IL-1β (**A**), IL-6 (**B**), TNF-α (**C**), TGF-β (**D**), and IL-10 (**E**). Untreated medium (Control) was used as a negative control and 1 µg/mL LPS was used as a positive control. Data are presented as mean ± standard deviation from five independent experiments. Differences between macrophages treated with different conditions were considered at *p* < 0.05, according to the analysis of variance (ANOVA) followed by a non-parametric test. * *p*-value < 0.05, ** *p*-value < 0.01, **** *p*-value < 0.0001. IL: interleukin, LPS: lipopolysaccharide, TGF: transforming growth factor, TNF: tumor necrosis factor.

## Data Availability

The original contributions presented in this study are included in the article/Appendix A. Further inquiries can be directed to the corresponding author.

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
