# Peer review of "Physalia physalis—A Source of Bioactive Collagen for the Cosmetic Industry"

_ijms, 2025, doi:10.3390/ijms27010033_

Round 1
Reviewer 1 Report
Comments and Suggestions for Authors
The article presents research on the extraction, characteristics, and bioactivity of collagen and collagen peptides from Physalia physalis. The work is extensive, well-structured, and covers a wide range of analyses: chemical, physicochemical, biological, and application-related. I found the work extremely interesting, the topic itself seems “timely” to me, the language and editorial preparation of the work are rather flawless. The paper is written in a logical way, the authors move freely to the following parts of the work.
Below I present comments/questions of concern that came to my mind while reading the reviewed paper. These are mainly due to the reviewer's curiosity in the area of the subject matter covered but some of them need to be discussed/reviewed:
- The CD showed a lack of the 220 nm band, suggesting collagen denaturation already at the extraction stage - can the authors indicate which steps of the protocol could potentially lead to denaturation? Were milder extraction conditions or optimization of time/temperature considered?
- Additional low-molecular-weight bands (<75 kDa) were observed in SDS-PAGE - have the authors confirmed that these are collagen degradation products?
- Why was “Hydroidolina subclass” used as a reference database instead of one created specifically for Physalia spp.? In my opinion, the limitations of the database used could have radically affected the discussion of the results.
- Did the authors perform additional tests to confirm proliferation, e.g., BrdU, Ki67, or cell counting (it seems to me that the resazurin reduction assay does not distinguish between proliferation and increased metabolic activity)?
- I’m puzzled by the high SD values for IL-10 results in stimulated macrophages - could the authors explain the reason for such high variability?
- Unfortunately, I think that the application section has been treated very superficially - this section is definitely lacking in data - at least the stability, rheology, and pH results should be included in the publication. Furthermore, could the authors explain why a W/O emulsion was chosen, given that many anti-aging products are O/W emulsions that facilitate application and sensory perception?
- I am also concerned about one issue, the explanation of which should be included in the manuscript - how does collagen denaturation affect the firmness, bioactivity, and cosmetic potential of the final cosmetic product?
Author Response
Reviewer 1
- The CD showed a lack of the 220 nm band, suggesting collagen denaturation already at the extraction stage - can the authors indicate which steps of the protocol could potentially lead to denaturation? Were milder extraction conditions or optimization of time/temperature considered?
Our reply: Thank you for your insightful question. In fact, several steps in the extraction protocol have the potential to destabilize the collagen triple helix. The alkaline pretreatment (0.1 M NaOH, 24 h), performed to remove non-collagenous materials, may damage the triple-helix structure if excessively prolonged; in our procedure, samples were stirred for 24 h. Mechanical homogenization using a food processor may also contribute to fibril fragmentation and chain exposure to acidic conditions. In addition, prolonged acidic extraction with glacial acetic acid under stirring can promote helix destabilization due to electrostatic repulsion and acid-induced unfolding.
The extraction procedure adopted in this study was selected based on previously published methods that reported high collagen extraction yields (doi:10.1016/j.afres.2025.101165; doi:10.3390/polym12102230). Therefore, alternative extraction conditions were not evaluated in this study. These methodological limitations were explained and incorporated into the revised manuscript.
“The extraction conditions used in this study, including the prolonged alkaline pre-treatment, mechanical shear during homogenization, and extended incubation with glacial acetic acid, which may promote electrostatic repulsion and acid-induced unfolding, could have contributed to structural damage of the collagen triple helix.” (page 7, lines 235-238).
“Future research should optimize extraction parameters to minimize structural damage and improve the preservation of native collagen architecture.” (page 21, lines 715-717).
- Additional low-molecular-weight bands (<75 kDa) were observed in SDS-PAGE - have the authors confirmed that these are collagen degradation products?
Our reply: Thank you for your comment. The SDS-PAGE profiles showed weak additional bands below 75 kDa. We did not experimentally confirm the identity of these bands. However, the description as possible collagen degradation products or smaller collagen fragments is consistent with published studies reporting that prolonged alkaline or acidic extraction conditions and mechanical shear can disrupt collagen structure, promoting partial cleavage of α-chains and generating lower-molecular-weight fragments that appear as faint bands in SDS-PAGE (doi:10.3390/polym12102230; doi:10.1016/j.lwt.2022.113439).
- Why was “Hydroidolina subclass” used as a reference database instead of one created specifically for Physalia spp.? In my opinion, the limitations of the database used could have radically affected the discussion of the results.
Our reply: Thank you for your comment. Hydroidolina subset was used (UniProt availability date 11/2025) because no curated Physalia physalis proteome was available at the time. This information was already explained in the manuscript: “For Physalia physalis, no collagen peptides were detected, suggesting either the absence of collagen sequences in the reference database used (which corresponded to Physalis, Hydroidolina subclass, and not Physalia physalis) or limited sequence homology pre-venting confident peptide assignment. This highlights the need for improved reference datasets for marine invertebrate species.” (page 7, lines 256-260).
- Did the authors perform additional tests to confirm proliferation, e.g., BrdU, Ki67, or cell counting (it seems to me that the resazurin reduction assay does not distinguish between proliferation and increased metabolic activity)?
Our reply: Thank you for your suggestion. As stated in the manuscript, the resazurin reduction assay was used solely to assess cellular metabolic activity: “The metabolic activity was measured using the resazurin reduction assay.” (page 21, lines 677–678).
Since no complementary assays were performed to directly evaluate cell proliferation, all the manuscript has been revised to ensure accurate interpretation of the results.
- I’m puzzled by the high SD values for IL-10 results in stimulated macrophages - could the authors explain the reason for such high variability?
Our reply: Thank you for your comment. We agree that the IL-10 data exhibit considerable variability. This variability may be attributed to biological heterogeneity among samples and the relatively small sample size, which can amplify variation. Additionally, differences between macrophage passages, batch-to-batch variation in LPS responsiveness, and technical variability associated with the ELISA assay (including sensitivity limits and plate-to-plate variation) may also have contributed to the dispersion in the IL-10 measurements.
We have included these considerations in the revised Discussion to contextualize the observed variability. “The high variability observed in IL-10 levels may reflect inherent biological heterogeneity combined with the small sample size, as well as technical variability associated with macrophage passage differences, batch-dependent LPS responsiveness, and ELISA sensitivity.” (page 15, lines 468-471).
- Unfortunately, I think that the application section has been treated very superficially – this section is definitely lacking in data - at least the stability, rheology, and pH results should be included in the publication. Furthermore, could the authors explain why a W/O emulsion was chosen, given that many anti-aging products are O/W emulsions that facilitate application and sensory perception?
Our reply: Thank you for your valuable suggestions. The stability, rheological and pH data were included in the manuscript as suggested:
“For the collagen biopeptide serum, formulation optimization ensured compatibility within pH 5.8 ± 0.2, the expected pH range for dermo cosmetic products. The serum matrix was structured using a polymeric base, temperature between 22–55°C, viscosity of 2800 ± 240 cP, and no phase separation during 30-day accelerated storage (4–45°C cycling), providing stability and desirable rheological properties” (page 17, lines 503-507).
Moreover, a W/O emulsion was used to encapsulate collagen biopeptides within an oily continuous phase. As already described in the manuscript, the selection of surfactants and oils with hydrophilic–lipophilic balance ≤ 9 favor stable W/O nanoemulsion formation, enhancing the protection and controlled release of collagen peptides, potentially improving skin penetration and bioavailability (please see page 17 lines 511-517).
- I am also concerned about one issue, the explanation of which should be included in the manuscript - how does collagen denaturation affect the firmness, bioactivity, and cosmetic potential of the final cosmetic product?
Our reply: Thank you for your question. Collagen denaturation can affect its mechanical strength and firmness of biomaterials, as intact fibrils can assemble into load-bearing networks. However, denatured collagen and low-molecular-weight hydrolyzed peptides retain significant bioactivity, as they are more readily absorbed by skin cells and can act as signaling molecules to stimulate fibroblast proliferation, modulate cytokine release, and promote extracellular matrix remodeling. In this study, the hydrolyzed collagen fractions demonstrated the ability to enhance cell migration and regulate macrophage cytokine production, supporting their potential role in regenerative cosmetic applications. Therefore, while denaturation may reduce the contribution to mechanical firmness, it does not compromise the bioactive and functional properties of the collagen for topical cosmetic use. This information was included in the manuscript (page 17, lines 523-532).
Reviewer 2 Report
Comments and Suggestions for Authors
Comments are shown in the attached file

Author Response
Reviewer 2
- The abstract does not adequately highlight the main findings. It should be revised to clearly present the key results, representative quantitative data, and the specific novelty of this study.
Our reply: Thank you for your suggestion. The abstract has been rewritten to emphasize the main quantitative findings and clearly highlight the novelty and outcomes of the study:
“Collagen, the most abundant structural protein in animals, is fundamental for tissue integrity and regeneration. Conventional mammalian sources face limitations related to sustainability, safety, and ethical concerns, underscoring the need for alternative bio-materials. Marine organisms, particularly jellyfish, offer a promising eco-friendly col-lagen source. In this study, collagen and collagen-derived peptides were extracted from the cnidarian Physalia physalis and biochemically characterized. Circular dichroism demonstrated partial loss of triple-helix structure, while SDS-PAGE revealed type I col-lagen related a-chains together with low-molecular-weight fragments. The hydrolyzed collagen fractions exhibited keratinocyte and fibroblast cytocompatibility and increased keratinocyte migration. Moreover, P. physalis-derived peptides modulated inflammatory cytokine release in lipopolysaccharide-stimulated macrophages reducing tumor necrosis factor (TNF)-a by 38% and increasing interleukin (IL)-10 by 29%. Based on these results, a stable bioactive serum formulation incorporating P. physalis collagen peptides was developed. Overall, this work demonstrates that bioactive peptides from P. physalis possess immunomodulatory and regenerative potential and represent a promising new marine resource for cosmetic applications.” (page 1, lines 20-35).
- Line 80-89. The scientific name should follow taxonomic conventions (first Physalia physalis, then P. physalis). Please also choose either “Portuguese man-o’-war” or the scientific name and use it consistently throughout the manuscript.
Our reply: Thank you for your correction. We have corrected all instances according to taxonomic rules.
- The Introduction does not clearly explain why Portuguese man-o’-war is a relevant candidate for cosmetic applications, nor does it summarize the current state ofjellyfish or Portuguese man-o’-war collagen or peptide use in this field. This background should be added.
Our reply: Thank you for your suggestion. This information was included in the introduction:
“Marine organisms have become valuable biological resources in the cosmetic in-dustry due to their richness in bioactive peptides and structural proteins that support skin regeneration and extracellular matrix remodeling (Song et al., 2023). Jellyfish-derived collagen has gained attention for its biocompatibility and antioxidant activity (Hu et al., 2025). P. physalis, a widely distributed cnidarian species, represents an abundant and under-explored source of biomolecules” (page 3, lines 94-99) and “Despite interest in jellyfish collagen, no previous research has investigated bioactive peptides derived from P. physalis for cosmetic use. Therefore, the aim of this study was to evaluate the characteristics of collagen and peptides extracted from P. physalis and to assess their biological potential for cosmetics” (page 3, lines 115-118).
- The rationale for selecting cod and haddock skin as comparator sources is not explained. Please justify this choice and clarify what type of benchmarking these controls are intended to provide.
Our reply: Thank you for your suggestion. This information was completed in Materials and methods, section 3.1: “Collagen extracted from cod and haddock skin was used as these species are widely used commercial marine collagen sources with well-described biochemical characteristics. Their inclusion provides a relevant performance baseline to evaluate the novelty and functional quality of P. physalis collagen.” (page 18, lines 543-546).
- The information in Lines 114—117 fits better in the Introduction or Materials and Methods.
Our reply: Thank you for your suggestion. This information was relocated and adjusted as in topic 3.
- The method used to determine collagen concentration in Section 2.1 is not described and must be specified. In addition, the results should be converted to units (e.g., mg/g dry weight) that allow direct comparison with published jellyfish collagen data.
Our reply: Thank you for your suggestions. The method of collagen determination was added as suggested “Collagen concentration was quantified using a hydroxyproline colorimetric assay with L-hydroxyproline standards (0–100 µg/mL, Sigma-Aldrich). Tissue collagen content was calculated using a conversion factor of 7.46 based on the average hydroxyproline composition of collagen” (page 18, lines 575-578).
Moreover, the results of collagen concentration were corrected to mg/g dry weight as suggested (Figure 1- page 3 and 4, lines 133-154).
- The literature-style paragraph in Lines 161-177 is more appropriate for the Introduction than for the Results and should be relocated.
Our reply: Thank you for your suggestion. These phrases were relocated at the introduction section:
Among them, type I collagen is predominant in skin, tendon, bone, and ligaments, where it provides tensile strength, and forms heterotrimeric fibrils composed of two a1(I) and one a2(I) chains (Amirrah et al., 2022). Type II collagen, in contrast, is a homotrimer of three a1(II) chains and is primarily found in cartilage and the vitreous humor of the eye. Type III collagen, often co-distributed with type I, contributes to the elasticity and repair ca-pacity of tissues such as skin, blood vessels, and internal organs (Ricard-Blum, 2011). Other fibrillar and non-fibrillar collagens, including type IV, IX, XII, and XIV, play specialized structural and signaling roles within the extracellular matrix (Shahrajabian and Sun, 2024). The structural and functional diversity of collagens is largely determined by their amino acid composition, particularly the content of proline and hydroxyproline, which stabilizes the triple-helical structure through steric constraints of the pyrrolidine ring and the formation of additional hydrogen bonds (Carvalho et al., 2018). These differences in molecular architecture and stability influence fibrillogenesis, cross-linking, and thermal resistance, thereby affecting both the biological role of each collagen type and its behavior during extraction and processing (James et al., 2023). This structural diversity underlies the wide-ranging biological roles of collagen and explains its versatility as a biomaterial.” (page 2, lines 55-71).
- Line 201-213. The FTIR discussion only lists characteristic bands and lacks deeper interpretation. Please discuss differences in band intensity or shifts between samples and derive clear structural conclusions from the spectra.
Our reply: Thank you for your comment. This information was included as suggested: “The ATR-FTIR spectra of P. physalis collagen exhibited the characteristic amide bands of collagen, including amide A (~3290–3310 cm⁻¹), amide B (~2925 cm⁻¹), amide I (1652–1638 cm⁻¹), amide II (1544–1532 cm⁻¹), and amide III (1240–1235 cm⁻¹). Compared with cod and haddock collagen controls, P. physalis displayed a noticeable reduction in amide I band intensity and a slight shift from 1652 to 1645 cm⁻¹, indicating decreased hydrogen bonding organization and partial disruption of the triple-helix structure. The amide III/Amide I absorbance ratio was lower for P. physalis (0.62) compared with cod collagen (0.86), also suggesting increased denaturation.” (page 6, lines 188-199).
- Line 214-288. The SDS-PAGE gel in Figure 2D does not clearly show the bands described in the text, and type I collagen bands are difficult to identify. The data are not convincing in their current form; repetition of the SDS-PAGE with improved resolution and a revised description is recommended.
Our reply: We thank the reviewer for this important comment and fully acknowledge the limitations in resolution and contrast of the SDS-PAGE image presented in Figure 2D. Unfortunately, the experiment cannot be repeated at this stage because the available collagen extract material has been fully used during the subsequent biological assays, enzymatic hydrolysis, and analytical characterizations (CD, FTIR, LC-MS, and cell experiments), and new biological material from Physalia physalis cannot currently be obtained. P. physalis is a seasonally available species whose collection is strictly regulated for safety and environmental reasons, and no additional fresh biomass is presently available for re-extraction.
- Figures need more standardized formatting. Each panel should include clear legends, axis labels, units, and treatment descriptions, and the overall style should be consistent across figures.
Our reply: Thank you for your suggestion. The images were corrected and replaced as suggested.
- The experimental work is relatively simple and focuses mainly on basic immunomodulatory assays. As such, the strong claims regarding “wound-healing and regenerative cosmetic formulations” are not fully supported and should either be toned down or backed by additional functional data.
Our reply: Thank you for your suggestion that were considered and improved along the manuscript.
- Line 457-486. The section on “Formulation of a bioactive serum” is purely descriptive and lacks supporting experimental data (e.g., stability, physicochemical characterization, activity). In its current form it is not acceptable.
Our reply: Thank you for your comment. All the information regarding pH, viscosity, and stability provided by Mesosystem S.A. was included in the paragraph:
“For the collagen biopeptide serum, formulation optimization ensured compatibility within pH 5.8 ± 0.2, the expected pH range for dermo cosmetic products. The serum matrix was structured using a polymeric base, temperature between 22–55°C, viscosity of 2800 ± 240 cP, and no phase separation during 30-day accelerated storage (4–45°C cycling), providing stability and desirable rheological properties” (page 17, lines 503-507).
- The Materials and Methods section is incomplete regarding sources and specifications of reagents and equipment. Please provide supplier and model information for all key instruments and kits (e.g., ATR-FTIR, LC-MS, Mini-PROTEAN system, ELISA kits, enzymes).
Our reply: Thank you for your suggestions that were already added in material and method section.“Commercial collagenase 1:6 (w/w) (LS0004180, PAN Biotech), papain 1:6 (w/w) (P4762, Sigma-Aldrich), and alcalase 1:6 (w/w) enzyme-to-substrate ratios (126741, Sigma-Aldrich) (10 mg; final enzyme concentration 1 mg/mL) were added to test the most efficient enzyme” (page 19, lines 581-584).
“Attenuated Total Reflection–Fourier Transform Infrared Spectroscopy (ALPHA II-Bruker compact FTIR spectrometer, Ettlingen, Germany)” (page 19, lines 591, 592).
“The peptides identified as collagen from the Mass Spectrometry (X500B positive Q-TOF MS, twin-spray ion source (Sciex))” (page 19, lines 621-622).
“Collagen and hydrolysates were resolved on 12% polyacrylamide gels at 110 V for 60 min using a Mini-PROTEAN Tetra Cell electrophoresis system (catalog no. 165-8030, Bio-Rad, Her-cules, CA, USA.).” (page 20, lines 631-633).
“The concentrations of tumor necrosis factor (TNF)-a (# 88-7324-88, Invitrogen), in-terleukin (IL)-1b (# 88-7013A-88, Invitrogen), IL-6 (# 88-7064-88, Invitrogen), IL-10 (# 88-7105-88, Invitrogen) and transforming growth factor (TGF)-b1 (# 88-8350-88, Invitro-gen) in cell culture supernatants were quantified using the corresponding Mouse Un-coated ELISA kit, following the manufacturer’s instructions.” (page 20, lines 657-661).
- Several reagent concentrations are ambiguously described (e.g., glacial/acetic acid, NaC1 solutions). All buffers and reagents should be reported with explicit concentrations to ensure reproducibility.
Our reply: Thank you for your suggestions that were already added in material and method section: “P. physalis was first washed with 0.1 M sodium hydroxide (NaOH) (1:10, w/v) under stirring for 24 h. The NaOH solution was then removed and discarded, and the samples were rinsed thoroughly with cold distilled water (Milli-Q Water Purification System, Germany) until the pH stabilized at approximately 7.0. The tissues were subsequently homogenized using a food processor (Yämmi Robot), and the homogenate was solubilized in 0.5 M glacial acetic acid (695092, Sigma-Aldrich, 99.8% purity). Specifically, 100 g of homogenized P. physalis tissue was mixed with 500 mL of distilled water and 2.5 mL of 0.5 M glacial acetic acid, followed by continuous stirring at 10 °C for at least 24 h. The resulting solution was centrifuged at 7500 rpm for 45 min at 4 °C to separate the supernatant from the pellet. The supernatant was neutralized to a pH of approximately 6.0 using concen-trated NaOH (43617, Sigma-Aldrich), and collagen was precipitated with 2 M sodium chloride (NaCl) (S9888, Sigma-Aldrich). The mixture was incubated for 1 h and centri-fuged again in the same conditions. The pellet was collected, washed twice with NaCl, and finally rinsed with distilled water. The pellet was then dissolved in 0.5 M acetic acid, supplemented with 7 mg pepsin (107192, Sigma-Aldrich), and stirred for 24 h. The pellet was then solubilized with 0.5 M acetic acid for three times, followed by a second pre-cipitation step with 0.9 M NaCl on ice. After resting for 1 h, the precipitate was centri-fuged, washed twice with NaCl, and a gradual dialysis with gradual 0.1 M, 0.05 M, and 0.025 M acetic acid was performed” (page 18, lines 548-566).
- Line 495. The phrase “Collagen extraction was optimized by the partners involved in this project” is too vague for a scientific paper. Please replace it with a concise, objective description of what was optimized, or remove it.
Our reply: Thank you for your comment. The phrase was removed as suggested.
- In Section 3.2 The overall temperature conditions during collagen extraction are not clearly reported. Given collagen's sensitivity to heat, all temperatures used at each step should be specified, and the authors should comment on how integrity was preserved.
Our reply: Thank you for your suggestion. This information was included in section 3.2: “All the steps were performed at 10 °C to avoid collagen degradation.” (page 18, lines 572, 573).
- Line 521. The activities of collagenase, papain, and alcalase and the enzyme-to-substrate ratios are not provided. These parameters should be reported to allow assessment and reproduction of the hydrolysis conditions.
Our reply: Thank you for your suggestions. The explanation of the use of the enzymes is provided in page 8, lines 293-295 “next, alcalase, collagenase, and papain, commonly used proteolytic enzymes, were evaluated for their ability to hydrolyze collagen from different sources into low-molecular-weight peptides”. Moreover, the parameters and enzyme-to-substrate ratios were included: “Commercial collagenase 1:6 (w/w) (LS0004180, PAN Biotech), papain 1:6 (w/w) (P4762, Sigma-Aldrich), and alcalase 1:6 (w/w) enzyme-to-substrate ratios (126741, Sigma-Aldrich) (10 mg; final enzyme concentration 1 mg/mL) were added to test the most efficient enzyme.” (page 19, lines 581-584).
- Line 562. The reference to Bos taurus is not explained. Please clarify what Bos taurus represents in the study and why this comparison is relevant.
Our reply: Thank you for your question. We clarified that Bos taurus collagen was included as a commercial reference standard commonly used in cosmetics. This justification now appears at page 19, lines 617, 618.
- Section 3.5. It is unclear whether all polyacrylamide gels used for collagen and peptide analysis were 12% or if different percentages were employed. Please specify gel compositions for each assay.
Our reply: Thank you for your suggestion. The concentration of polyacrylamide gels used for collagen and hydrolysates was the same (12%) . This information was added at page 20, line 631.
- The references should be carefully revised to conform to the journal's formatting requirements.
Our reply: Thank you for your suggestion. All the references were revised.
Reviewer 3 Report
Comments and Suggestions for Authors
This manuscript presents a comprehensive and well-structured study on the extraction, characterization, and bioactivity evaluation of collagen and collagen-derived peptides from Physalia physalis (Portuguese man-o'-war). The research addresses a relevant need for sustainable, non-mammalian collagen sources for the cosmetic industry. The work is innovative, linking the valorization of an underutilized and problematic marine organism to high-value biotechnological applications. The experimental design is robust, incorporating comparisons with established marine sources and a commercial sample. The collaboration with an industrial partner for formulation development strengthens the translational impact of the findings.
The manuscript is generally well-written, but it requires significant revisions to enhance clarity, data interpretation, and methodological detail before it can be considered for publication. The major concerns revolve around the characterization data (particularly the denatured state of the extracted collagen and the absence of Physalia peptides in MS analysis), the need for more precise statistical reporting in figures, and a more critical discussion that acknowledges the limitations of the study. The following section-by-section review provides detailed comments and suggestions for improvement.
Major
- The transition from marine collagen in general to Physalia physalis could be slightly smoother. A sentence explicitly stating "Among these marine sources, jellyfish and related cnidarians have shown promise, and Physalia physalis represents a particularly interesting candidate due to..." would improve the flow.
- The authors should convert their data to a dry weight basis (if the dry weight of the starting material is known) to allow for a more direct and meaningful comparison with the cited literature. If not possible, this should be explicitly stated as a limitation in the discussion.The authors must discuss the implications of extracting denatured collagen (gelatin). They should address:
- a) Why the extraction process led to denaturation (e.g., acidic conditions, pepsin treatment). b) How this affects the interpretation of subsequent bioactivity results, as the activities are likely due to gelatin and its peptides, not native collagen. The title and text should be precise; it is primarily "gelatin" and "gelatin-derived peptides" that are being studied.
3- Briefly mention the specific activity or units of the enzymes used, as concentration (mg/mL) alone can be misleading if enzyme specific activities differ.
- CD spectroscopy indicates a denatured structure, making subtype classification uncertain.Mass spectrometry results failed to detect Physalia-specific collagen peptides; this limitation should be addressed comprehensively. Provide stronger structural evidence OR tone down the claim and classify it simply as “collagen-derived material”.
- The extraction workflow lacks essential information for reproducibility.No yield percentage reported.No justification for selected buffer conditions.The process should be illustrated in a schematic diagram. Expand the Methods section with complete steps, yields, and rationale.
- Some extracts increased inflammatory cytokines in non-stimulated macrophages.This contradicts claims of anti-inflammatory activity.Possible explanations must be discussed (endotoxin contamination, concentration effects, peptide nature, etc.).Provide a full discussion and avoid overgeneralized claims.
- Ensure all figure labels are fully visible. Statistically significant differences should be clearly marked between each test group and the control at each concentration, not just within a group across concentrations.
- The description in the text is qualitative ("comparable and markedly higher," "complete wound closure"). Quantitative data (e.g., percentage of wound closure at 6h and 24h) is essential and should be presented in a graph alongside the representative images.
- The figures (6A-J) are complex and would benefit from a clearer visual design to help the reader distinguish between the numerous groups and cytokines.Consider splitting Figure 6 into two figures (Naive vs. LPS-stimulated) for clarity. Ensure all statistical comparisons are explicitly described in the figure legend and text.
- The manuscript lacks: Correction for multiple comparisons.Effect size reporting.Justification for small sample sizes (n=3, N=5).Improve statistical rigor or acknowledge the limitations clearly.
- All figures need more detailed captions that fully describe what is being shown without forcing the reader to refer back to the main text. Define all abbreviations in the captions.
- The current conclusions claim “validation” of Physalia as a collagen source, but the evidence is not fully conclusive.Rephrase to moderate, evidence-based claims.
The manuscript is acceptable for publication pending satisfactory revisions as requested by the reviewers.
Author Response
Reviewer 3
- The transition from marine collagen in general to Physalia physalis could be slightly smoother. A sentence explicitly stating "Among these marine sources, jellyfish and related cnidarians have shown promise, and Physalia physalis represents a particularly interesting candidate due to..." would improve the flow.
Our reply: Thank you for your suggestion that was taken in consideration in the introduction section: “Marine organisms have become valuable biological resources in the cosmetic in-dustry due to their richness in bioactive peptides and structural proteins that support skin regeneration and extracellular matrix remodeling (Song et al., 2023). Jelly-fish-derived collagen has gained attention for its biocompatibility and antioxidant activity (Hu et al., 2025). P. physalis, a widely distributed cnidarian species, represents an abundant and under explored source of biomolecules..” (page 3, lines 94-99).
- The authors should convert their data to a dry weight basis (if the dry weight of the starting material is known) to allow for a more direct and meaningful comparison with the cited literature. If not possible, this should be explicitly stated as a limitation in the discussion.
Our reply: Thank you for your comment. The results of collagen extraction were converted to mg/g dry weight to allow comparison with the cited literature (page 3 and 4, lines 133-154).
- The authors must discuss the implications of extracting denatured collagen (gelatin). They should address: a) Why the extraction process led to denaturation (e.g., acidic conditions, pepsin treatment). b) How this affects the interpretation of subsequent bioactivity results, as the activities are likely due to gelatin and its peptides, not native collagen. The title and text should be precise; it is primarily "gelatin" and "gelatin-derived peptides" that are being studied.
Our reply: We thank the reviewer for this important and constructive observation. We agree that our analytical data demonstrate that the collagen extracted has a degree of denaturation. We included in the discussion section the steps of collagen extraction that can impact collagen stability:
“The extraction conditions used in this study, including the prolonged alkaline pre-treatment, mechanical shear during homogenization, and extended incubation with glacial acetic acid, which may promote electrostatic repulsion and acid-induced unfolding, could have contributed to structural damage of the collagen triple helix” (page 7, lines 235-238).
This collagen denaturation can affect the fibrillar organization and mechanical reinforcement properties typically attributed to native triple-helical collagen. This may limit the contribution of the extracted material to structural firmness or scaffold-like mechanical reinforcement in final formulations. However, as our current work focuses on biological functionality rather than mechanical performance, no direct mechanical testing within the scope of this study was performed. This information is included at page 17, lines 530-532: “Therefore, while denaturation may reduce the contribution to mechanical firmness, it does not compromise the bioactive and functional properties of the collagen for topical cosmetic use”.
As the extracted material is unambiguously collagen-derived and enzymatically converted from collagen, which is consistent with common practice in the biomaterials and cosmetic literature, we believe that maintaining the term “collagen” in the title remains scientifically appropriate.
4: Briefly mention the specific activity or units of the enzymes used, as concentration (mg/mL) alone can be misleading if enzyme specific activities differ.
Our reply: Thank you for the suggestion. The information was completed in section 3.4: “The extracted collagen (60 mg) was resuspended in 10 mL of phosphate-buffered saline (PBS 1×, pH 7.4; PAN Biotech). Commercial collagenase (LS0004180, PAN Biotech), papain (P4762, Sigma-Aldrich), and alcalase (126741, Sigma-Aldrich) (10 mg; final en-zyme concentration 1 mg/mL) 1:6 (w/w) enzyme-to-substrate ratios were added to test the most efficient enzyme.” (page 19, lines 580-584).
5: CD spectroscopy indicates a denatured structure, making subtype classification uncertain. Mass spectrometry results failed to detect Physalia-specific collagen peptides; this limitation should be addressed comprehensively. Provide stronger structural evidence OR tone down the claim and classify it simply as “collagen-derived material”.
Our reply: Thank you for your question. Mass spectrometry results are related to collagen samples before enzyme treatment.
6: The extraction workflow lacks essential information for reproducibility.No yield percentage reported.No justification for selected buffer conditions.The process should be illustrated in a schematic diagram. Expand the Methods section with complete steps, yields, and rationale.
Our reply: Thank you for your suggestion. This section was improved and a schematic diagram was included as supplementary figure 1: “Specifically, 100 g of homogenized P. physalis tissue was mixed with 500 mL of distilled water and 2.5 mL of 0.5 M glacial acetic acid, followed by continuous stirring at 10 °C for at least 24 h. The resulting solution was centrifuged at 7500 rpm for 45 min at 4 °C to separate the supernatant from the pellet. The supernatant was neutralized to a pH of approximately 6.0 using concentrated NaOH (43617, Sigma-Aldrich), and collagen was precipitated with 2 M sodium chloride (NaCl) (S9888, Sigma-Aldrich). The mixture was incubated for 1 h and centrifuged again in the same conditions. The pellet was collected, washed twice with NaCl, and finally rinsed with distilled water. The pellet was then dissolved in 0.5 M acetic acid, supplemented with 7 mg pepsin (107192, Sigma-Aldrich), and stirred for 24 h. The pellet was then solubilized with 0.5 M acetic acid for three times, followed by a second precipitation step with 0.9 M NaCl on ice. After resting for 1 h, the precipitate was centrifuged, washed twice with NaCl, and a gradual dialysis with gradual 0.1 M, 0.05 M, and 0.025 M acetic acid was performed. The extracted collagen was sub-sequently lyophilized and stored under appropriate conditions until further use.” (page 18, lines 553-567).
7: Some extracts increased inflammatory cytokines in non-stimulated macrophages.This contradicts claims of anti-inflammatory activity.Possible explanations must be discussed (endotoxin contamination, concentration effects, peptide nature, etc.).Provide a full discussion and avoid overgeneralized claims.
Our reply: Thank you for this important comment. We have now expanded the discussion to address the observed increase in pro-inflammatory cytokines in naïve macrophages and to avoid any overgeneralized statements regarding anti-inflammatory activity.
In the revised manuscript (Section 2.4.3. Macrophage-derived inflammatory cytokines, page 13, lines 430-443), we added two mechanistic explanations supported by the literature. First, collagen fragments may act as danger-associated molecular patterns (DAMPs), capable of activating TLR2/TLR4 pathways in macrophages even in the absence of LPS, as previously described for several collagen-derived peptides (Thomas et al., 2007). This provides a plausible biological basis for the mild cytokine induction observed with the commercial hydrolysate.
Second, we discuss the possibility that differences in molecular-weight distribution, hydrolysate heterogeneity, and residual processing components—which are known to vary widely between commercial collagen preparations—may contribute to stronger basal macrophage activation when compared with the more homogeneous marine-derived peptides. We clarified that this is a hypothesis, consistent with known variability in commercial collagen hydrolysates, and not a definitive conclusion.
Importantly, we now explicitly state that no pro-inflammatory basal activation was observed with Physalia physalis, cod skin, or haddock peptides, which instead demonstrated a favourable immunomodulatory pattern. We also added a statement emphasizing that these results do not validate universal anti-inflammatory activity, but rather support a source-dependent immunomodulatory profile.
8: Ensure all figure labels are fully visible. Statistically significant differences should be clearly marked between each test group and the control at each concentration, not just within a group across concentrations.
Our reply: Thank you for your comment, all the figures were revised.
9: The description in the text is qualitative ("comparable and markedly higher," "complete wound closure"). Quantitative data (e.g., percentage of wound closure at 6h and 24h) is essential and should be presented in a graph alongside the representative images.
Our reply: Thank you for your question. A new graph with wound closure (%) was added and discussed:
“At the initial time point (0 h), a clear wound gap was observed in all groups. Pro-gressive closure of the scratch was evident over time, with cell migration becoming prominent after 6 h and nearly complete by 24 h in most treatments (Figure 5A). At 6 h, the control group exhibited minimal wound closure (5.17 ± 1.72%), and the commercial peptide group showed a similar low closure (3.89 ± 1.68%). Cells treated with P. physa-lis-derived peptides showed a comparable closure to the control (5.14 ± 0.96%), whereas cod skin- and haddock-derived peptides markedly enhanced migration, with wound closure of 13.74 ± 0.47% and 18.93 ± 0.82%, respectively. By 24 h, wound closure had progressed substantially in all groups. However, all the experimental groups presented significantly higher wound closure compare to the control group (80.46 ± 1.70%). Treatments with commercial, P. physalis, cod skin, and haddock peptides achieved nearly complete closure, with values of 85.11 ± 1.58%, 93.08 ± 0.63%, 98.10 ± 1.90%, and 95.06 ± 0.82%, respectively (Figure 5B). Cod skin peptides demonstrated the highest ef-ficacy, promoting almost complete wound closure within 24 h.” (page 11-12, lines 377-400).
10: The figures (6A-J) are complex and would benefit from a clearer visual design to help the reader distinguish between the numerous groups and cytokines.Consider splitting Figure 6 into two figures (Naive vs. LPS-stimulated) for clarity. Ensure all statistical comparisons are explicitly described in the figure legend and text.
Our reply: Thank you for your suggestion sthat were considered- Now Figure 6 represent naïve macrophage conditions and Figure 7 LPS-stimulated macrophages (page 14-16)
11: The manuscript lacks: Correction for multiple comparisons.Effect size reporting.Justification for small sample sizes (n=3, N=5).Improve statistical rigor or acknowledge the limitations clearly.
Our reply: Thank you for your suggestion that were revised in all the figures of the manuscript and in section 3.9: “Statistical comparisons were analyzed in SPSS Statistics, version 27.0.1.0 (IBM SPSS Statistics Software, Chicago, IL, USA), and GraphPad Prism, version 8.4.0 (GraphPad Software, San Diego, CA, USA). Normality was measured by the Shapiro-Wilk statistical test and was assumed when p > 0.05. All data were evaluated by a one-way analysis of variance (ANOVA). When statistical differences were identified, the variables were compared using Tukey’s multiple range test. As most datasets did not meet normal distribution, were performed using the Wilcoxon test with Bonferroni correction. When homogeneity of variances was not applied, Welch test followed by a post-hoc Games-Howell test was performed. Results were presented as mean ± standard deviation (SD) and the level of statistical significance was considered at p < 0.05.” (page 21, lines 698-707).
12: All figures need more detailed captions that fully describe what is being shown without forcing the reader to refer back to the main text. Define all abbreviations in the captions.
Our reply: Thank you for your question. All the figures were revised and improved as suggested.
13: The current conclusions claim “validation” of Physalia as a collagen source, but the evidence is not fully conclusive. Rephrase to moderate, evidence-based claims.
Our reply: Thank you for your suggestion. The conclusion section was improved accordingly: “This study demonstrated P. physalis as a novel and sustainable source of marine collagen with distinct structural and biological properties for cosmetic application. The extracted collagen and derived peptides exhibited high yield and biocompatibility, promoting keratinocyte migration, fibroblast activity, and macrophage-mediated an-ti-inflammatory responses. The successful development of a bioactive serum formulation in collaboration with Mesosystem (Porto, Portugal) highlights the translational potential of this marine-derived biomaterial for dermo cosmetic applications. Future research should optimize extraction parameters to minimize structural damage and improve the preservation of native collagen architecture, as well as, explore the molecular mechanisms underlying the bioactivity of P. physalis peptides, optimize large-scale extraction and purification methods, and evaluate their in vivo efficacy and safety in skin regeneration and wound-healing models. Developing these investigations could consolidate P. physalis as a sustainable and high-value collagen source within the blue biotechnology sector.” (page 21, lines 709-721).
Round 2
Reviewer 1 Report
Comments and Suggestions for Authors
Thank you to the Authors for the effort they put into addressing my comments and concerns.
All comments have been fully addressed and clarified. I am satisfied with the authors’ revisions and explanations. In my opinion, the manuscript is suitable for publication in its current form.
Author Response
We sincerely thank Reviewer 1 for the constructive and insightful suggestions. The comments were extremely helpful and contributed significantly to improving the clarity, structure, and overall quality of the manuscript.
Reviewer 2 Report
Comments and Suggestions for Authors
d
The author has well refined the manuscript according to the comments. But there are still some problems that could be solved to improve the manuscript's quality.
- Athough author has added some date in "2.5. Formulation of a bioactive serum enriched with collagen hydrolysates" to support the conclusion, the figures or tables, even the figures of production are suggested to added in the main text or in the supplementary materials to support the results like "no phase seperation". If it is impossible to provide, please demonstrate the appropriate reason.
- Author has metioned that "collagen and peptide analysis were used in 12% seperating gel". However, the SDS-PAGE of collagen peptides cannot be found in the main text, please provide the correlative figure.
Author Response
Dear Editor,
In reply to the review performed on the paper entitled “Red grape stem varieties as a source of tryptophan and selenium: Functional properties and antioxidant potential”, we would like to acknowledge the valuable minor revisions performed by reviewer 2. We hope the answers below and modifications that have been done in the manuscript are clear and concise enough as required to enable the publication of the manuscript.
Answer to referee’s comments and queries
Reviewer 2
- Athough author has added some date in "2.5. Formulation of a bioactive serum enriched with collagen hydrolysates" to support the conclusion, the figures or tables, even the figures of production are suggested to added in the main text or in the supplementary materials to support the results like "no phase seperation". If it is impossible to provide, please demonstrate the appropriate reason.
Our reply: Thank you for your insightful comment. In response to your suggestion, we have now added further information to Section 2.5 providing a detailed explanation of the phase stability observed in both serum formulations:
“This stability is attributed to the optimized polymeric network, which provides adequate structuring to entrap the aqueous phase and evenly disperse the collagen biopeptides. The selected polymer creates strong intermolecular interactions that prevent water mi-gration and maintain a stable gel-like matrix, ensuring homogeneity throughout the storage period.” (Page 17, lines 510-515).
“The stability assessment included visual inspection and storage testing, both indicating no phase separation. In this prototype, phase stability results from the correct ratio of sur-factants, which provides sufficient interfacial tension reduction, and from the formation of a tight oil-continuous phase that restricts droplet coalescence or creaming, thereby maintaining the structural integrity of the nanoemulsion.” (Page 17, lines 522-527).
Some detailed production parameters remain confidential due to industrial intellectual property restrictions. However, all non-confidential information relevant to supporting the reported results was also included in Supplementary Figure 1 and 2 legends. We believe these additions fully address the reviewer’s request and improve the transparency and robustness of the manuscript:
“The figure illustrates the formulation workflow, including dissolution of collagen hydrolysates in the aqueous phase, incorporation into the polymeric matrix, pH adjustment, and viscosity optimization. During a 30-day accelerated stability test (4–45 °C cycling), no phase separation was observed, and the serum maintained a homogeneous appearance and consistent rheological properties, indicating the formation of an adequate and stable polymeric network.” (Supplementary Figures document, lines 6-10).
“Visual stability assessments confirmed that the nanoemulsion maintained its structural integrity (at 4–45 °C cycling) and exhibited no phase separation throughout the testing period, reflecting the effectiveness of the selected surfactant system and its optimized ratios.” (Supplementary Figures document, lines 13-15).
- Author has metioned that "collagen and peptide analysis were used in 12% seperating gel". However, the SDS-PAGE of collagen peptides cannot be found in the main text, please provide the correlative figure.
Our reply: Thank you for your correction. The SDS-PAGE was only performed for collagen. The information was accordingly corrected at section 3.6 of the materials and methods: “Sodium dodecyl sulfate polyacrylamide gel electrophoresis (SDS-PAGE) was per-formed to evaluate the protein profile of the extracted collagen. Briefly, 20 µL of each sample was mixed with 5 µL of loading buffer (Nzytech) and denatured at 95 °C for 5 min. Collagens were resolved on 12% polyacrylamide gels at 110 V for 60 min using a Mini-PROTEAN Tetra Cell electrophoresis system” (page 20, lines 638-642).

Reviewer 3 Report
Comments and Suggestions for Authors
The manuscript has been reviewed and the necessary revisions have been made; therefore, it has been approved
Author Response
We are very grateful to Reviewer 3 for the careful evaluation and valuable recommendations. The suggestions provided were instrumental in strengthening the manuscript and enhancing its scientific rigour.